# Changes in the composition of marine and sea-ice diatoms derived from sedimentary ancient DNA of the eastern Fram Strait over the past 30,000 years

Heike H. Zimmermann[1], Kathleen R. Stoof-Leichsenring[1], Stefan Kruse[1], Juliane Müller[2,3,4], Ruediger Stein[2,3,4], Ralf Tiedemann[2], Ulrike Herzschuh[1,5,6]

[1]Polar Terrestrial Environmental Systems, Alfred Wegener Institute Helmholtz Centre for Polar and Marine Research, Potsdam, 14473, Germany
[2]Marine Geology, Alfred Wegener Institute Helmholtz Centre for Polar and Marine Research, 27568 Bremerhaven, Germany
[3]MARUM, University of Bremen, 28359 Bremen, Germany
[4]Faculty of Geosciences, University of Bremen, 28334 Bremen, Germany
[5]Institute of Biochemistry and Biology, University of Potsdam, 14476 Potsdam, Germany
[6]Institute of Environmental Sciences and Geography, University of Potsdam, 14476 Potsdam, Germany

*Correspondence to*: Heike H. Zimmermann (heike.zimmermann@awi.de)

**Abstract.** The Fram Strait is an area with a relatively low and irregular distribution of diatom microfossils in surface sediments, and thus microfossil records are scarce, rarely exceed the Holocene and contain sparse information about past richness and taxonomic composition. These attributes make the Fram Strait an ideal study site to test the utility of sedimentary ancient DNA (*sed*aDNA) metabarcoding. Amplifying a short, partial *rbcL* marker from samples of sediment core MSM05/5-712-2, resulted in 95.7 % of our sequences being assigned to diatoms across 18 different families with 38.6 % of them being resolved to species and 25.8 % to genus level. Independent replicates show high similarity of PCR products, especially in the oldest samples. Diatom *sed*aDNA richness is highest in the Late Weichselian and lowest in Mid- and Late-Holocene samples. Taxonomic composition is dominated by cold-water and sea-ice associated diatoms and suggests several re-organizations – after the Last Glacial Maximum, after the Younger Dryas, after the Early and after the Mid-Holocene. Different sequences assigned to, amongst others, *Chaetoceros socialis* indicate the detectability of intra-specific diversity using *sed*aDNA. We detect no clear pattern between our diatom *sed*aDNA record and the previously published IP$_{25}$ record of this core, although proportions of pennate diatoms increase with higher IP$_{25}$ concentrations and proportions of *Nitzschia* cf. *frigida* exceeding 2 % of the assemblage point towards past sea-ice presence.

## 1 Introduction

The marine environment is a complex ecosystem in which the distribution of organisms is controlled significantly by abiotic constraints such as sea-surface temperatures (SSTs), salinity, nutrient distribution, light conditions, and sea-ice cover (Cherkasheva et al., 2014; Ibarbalz et al., 2019; Nöthig et al., 2015; Pierella Karlusich et al., 2020). Over the past 30,000 years the subarctic North Atlantic Ocean was subject to frequent sea-ice expansions and contractions (Müller et al., 2009; Müller

and Stein, 2014; Syring et al., 2020; Werner et al., 2013), which are expected to have affected the composition of the regional species pool. Diatoms (Bacillariophyta) are unicellular, siliceous organisms that are photoautotrophic and thrive in the euphotic zone of the ocean. Owing to their sensitive responses to environmental conditions, diatoms are frequently used as indicators

for paleoenvironmental reconstructions, to assess, for example, changes in surface water temperatures (Birks and Koç, 2002; Krawczyk et al., 2017; Miettinen et al., 2015), paleoproductivity (Fahl and Stein, 1997; Limoges et al., 2018), and sea-ice distribution (Smirnova et al., 2015; Weckström et al., 2013). Next to microfossil-based reconstructions, the diatom produced sea-ice proxy $IP_{25}$ (a highly branched isoprenoid alkene with 25 carbon atoms; Belt et al., 2007) combined with phytoplankton biomarkers (e.g. brassicasterol, dinosterol; Volkman, 1986) permit semi-quantitative reconstructions of past sea-ice

distribution (Belt, 2018; Belt and Müller, 2013; Müller et al., 2009; Müller and Stein, 2014; Stein et al., 2012, 2017). However, diatoms in northern high-latitudinal regions are less silicified and more prone to silica dissolution (Kohly, 1998; Stabell, 1986) compared to diatoms of the southern polar oceans (Harrison and Cota, 1991). In the Fram Strait – an important area of heat exchange between Arctic and North Atlantic water masses (Untersteiner, 1988) – particularly low and irregular preservation of diatom microfossils prevails in surface sediments (Karpuz and Schrader, 1990; Stabell, 1987). The diatom records are

generally underrepresented, contain sparse information about past diversity and taxonomic composition, and rarely exceed the Holocene (Jessen et al., 2010; Koç et al., 2002; Stabell, 1986). This makes the Fram Strait an excellent site to test ancient DNA metabarcoding on a sediment core (Müller et al., 2012; Müller and Stein, 2014).

Ancient DNA is a new proxy that can exploit diatoms as indicators of past marine environmental change (Coolen et al., 2007; De Schepper et al., 2019; Kirkpatrick et al., 2016). Deep-sea sediments have been reported to be rich in DNA with up to 70–

50 90 % of the total DNA pool being extracellular DNA (Coolen et al., 2007; Dell'Anno et al., 2002; Lejzerowicz et al., 2013; Morard et al., 2017), and traces of DNA can be detected in sediments even though microfossils are absent or highly degraded (Boere et al., 2009; Coolen et al., 2009, 2013; Lejzerowicz et al., 2013; Pawłowska et al., 2020). Therefore, analyses of sedimentary ancient DNA (sedaDNA) could be advantageous in areas of biased preservation due to high silica dissolution rates. Beyond morphological or biogeochemical analyses, ancient DNA can distinguish cryptic species that are

55 morphologically similar (Stoof-Leichsenring et al., 2012), trace temporal changes of intra-specific genetic variation (Epp et al., 2018; Parducci et al., 2012; Zimmermann et al., 2017b), and identify genetic relationships and microevolution (Stoof-Leichsenring et al., 2014, 2015).

Paleogenetic analyses with an emphasis on diatoms have been successfully carried out in various limnic settings ranging from the subarctic (Epp et al., 2015; Stoof-Leichsenring et al., 2014, 2015) to the tropics (Stoof-Leichsenring et al., 2012) and

60 Antarctica (Coolen et al., 2004). Yet, it is still a relatively underrepresented branch in the marine realm with only a few published studies targeting phytoplankton ancient DNA (Boere et al., 2009, 2011b, 2011a; Coolen et al., 2006, 2007, 2009, 2013; De Schepper et al., 2019; Giosan et al., 2012; Kirkpatrick et al., 2016) or diatoms in particular (Pawłowska et al., 2020). In this study we examine ancient diatom DNA retrieved from sediments from the eastern Fram Strait and assess whether it can be used to analyze temporal changes in the taxonomic composition of diatoms. We have three principal objectives: (1) to assess

the quality and replicability of the data obtained by sedaDNA metabarcoding, (2) to analyze temporal changes of diatom

taxonomic composition and richness, and (3) to evaluate diatom *sed*aDNA as a new proxy for sea-ice reconstruction. DNA was derived from distinct samples of the comprehensively analyzed marine sediment core MSM05/5-712-2, covering the major climatic intervals since the Late Weichselian (i.e. the last ~30 kyr BP). As previous work indicates variability in the past sea-ice cover (Falardeau et al., 2018; Müller et al., 2012; Müller and Stein, 2014; Werner et al., 2011, 2013), samples were chosen according to high, medium, and low concentrations of the diatom produced sea-ice biomarker IP$_{25}$ (Müller et al., 2012; Müller and Stein, 2014) and we expect associated changes in the taxonomic composition. We used *sed*aDNA metabarcoding by applying the diatom-specific *rbcL_76* marker (Stoof-Leichsenring et al., 2012) which has already proved successful in low-productivity lakes of northern Siberia (Dulias et al., 2017; Stoof-Leichsenring et al., 2014, 2015), but so far has not been tested on marine sediments. The marker amplifies a short region of the *rbcL* gene, which is located on the chloroplast that is present in diatoms in several copy numbers (Vasselon et al., 2018), thereby increasing the probability of its long-term preservation. The *rbcL* gene has an adequate sequence reference database and was tested as a potential diatom barcode marker with high resolution power (Guo et al., 2015; Kermarrec et al., 2013). Furthermore, it reduces co-amplification of non-photosynthetic bacteria or archaea that are active in subsurface sediments and thus could be preferentially amplified during PCR in comparison to the highly fragmented and damaged ancient DNA.

## 2 Materials and Methods

### 2.1 Study site and sample material

The kastenlot core MSM05/5-712-2 (N 78.915662, E 6.767167, water depth 1487 m) was collected from the western continental slope of Svalbard during the cruise of *Maria S. Merian* (Budéus, 2007) in the eastern Fram Strait in August 2007 (Fig. 1). On board, subsections of 1 m length were placed in square plastic boxes as explained in the supplement of Gersonde et al. (2012) and stored at 4°C. This may have affected DNA preservation. The Fram Strait is located between Greenland and Svalbard and connects the Arctic Ocean with the Atlantic Ocean. The study area is influenced by temperate, saline water masses that are transported northwards via the West Spitzbergen Current, which is a continuation of the North Atlantic Current (Aagaard, 1982). Furthermore, the site is located downslope from Kongsfjorden and is thus influenced by one of the major outlets of western Svalbard meltwater (Werner et al., 2013). Today, the site is located south of the winter and summer sea-ice margin and is ice-free year round (Fig. 1). Age-depth modeling suggests a maximum age of about 30 cal kyr BP for the lowermost core interval (Müller and Stein, 2014). The sampling procedure for ancient DNA analyses followed the protocol for non-frozen sediment cores explained in Epp et al. (2019) for the 12 samples taken along the core, of which the depth at 8.85 m was sampled twice to check whether different samples from the same level show similar taxonomic composition and richness, particularly the oldest sample (Table 1).

## 2.2 DNA extraction, PCR, and sequencing

The DNA extractions and PCR setups were prepared in a dedicated laboratory for ancient DNA at the Alfred Wegener Institute, Helmholtz Centre for Polar and Marine Research (Potsdam, Germany). Total DNA was extracted from 12 samples of approximately 2 g (wet weight) sediment using the same method as described in Zimmermann et al. (2017a). Each extraction batch included one negative control. The DNA concentrations were measured with the Qubit dsDNA BR Assay Kit (Invitrogen, Carlsbad, CA, USA) on a Qubit 2.0 fluorometer (Invitrogen, Carlsbad, CA, USA). As DNA concentrations were below the detection limit, we concentrated 600 µL of each sample with the GeneJET PCR Purification KIT (Thermo Scientific, Carlsbad, CA, USA) according to the manufacturer's protocol and eluted twice with 15 µL elution buffer. All DNA extracts were stored at -20°C.

We amplified the marker *rbcl_76* (Stoof-Leichsenring et al., 2012), a 76 bp long fragment of the plastid *rbcL* gene using tagged primers *Diat_rbcL_705F* (AACAGGTGAAGTTAAAGGTTCATAYTT) and *Diat_rbcL_808R* (TGTAACCCATAACTAAATCGATCAT) as described in Dulias et al. (2017). The PCR reactions were set up in small batches, each including up to 9 samples and the corresponding negative control from the DNA extraction as well as a PCR no template control (NTC). For each sample, extraction negative control, and NTC, we performed 3 PCRs with different primer-tag combinations on different days. The PCR reaction mixes and conditions were prepared following the adjusted protocol for tagged *Diat_rbcL_705F* and *Diat_rbcL_808R* primers as described in Dulias et al. (2017) with the exception that 3 µl DNA (DNA concentration 3 ng µL$^{-1}$) was used as a template. PCRs were performed with the following settings: 5 minutes at 94°C (initial denaturation), then 50 cycles at 94°C (denaturation), 49°C (annealing), and 68°C (elongation), and a final elongation step at 72°C for 5 minutes. Subsequently the PCR success was checked with gel-electrophoresis. All PCR products were purified with the MinElute purification Kit (Qiagen, Hilden, Germany) according to the manufacturer's recommendations. Elution was carried out twice in 10 µL elution buffer. The purified PCR products were mixed in equal concentrations and sent to Fasteris SA sequencing service (Switzerland) who carried out library preparation and sequencing. The sequencing library was prepared with the Mid Output kit v. 2 according to the Fasteris Metafast protocol for low complexity amplicon sequencing and checked by qPCR. The library was sequenced (2 x 150 bp, paired-end) on the Illumina NextSeq 500 (Illumina Inc., San Diego, CA, USA). The sequence data are deposited in the Sequence Read Archive (BioProject: PRJxxxx).

## 2.3 Bioinformatic processing

The sequences were processed, filtered, and assigned a taxonomic name according to the NCBI taxonomy using the OBITools package (Boyer et al., 2016) with the same bioinformatics parameter settings as described in Dulias et al. (2017). We refrained from clustering sequences into operational taxonomic units and used amplicon sequence variants (ASVs) as recommended by Callahan et al. (2017). The OBITools program *obiclean* can identify ASVs that have likely arisen due to PCR or sequencing errors. It uses the information of sequence counts and sequence similarities to classify whether a sequence is a variant ("internal") of a more abundant ("head") ASV (Boyer et al., 2016). To generate the reference database for the taxonomic

assignment of the sequences we downloaded the EMBL release 138 (released November 2018) and used *ecopcr* (Ficetola et al., 2010) according to the descriptions of Dulias et al. (2017) and Stoof-Leichsenring et al. (2012) containing 2,320 reference

sequences. A total of 7,536,449 sequence counts were assigned to samples, 235 sequence counts to extraction negative controls and 237 counts to PCR negative controls. Of the 204 different sequence variants detected in extraction and PCR negative controls, 83% of their occurrences were singletons (Suppl. Table 1) and most likely artefacts from tag-jumping during library preparation (Schnell et al., 2015). Using R v. 3.5.0 (R Core Team, 2018), we kept only those ASVs (Suppl. Table 2) that (1) were assigned a taxonomic name based on 90–100 % similarity to an entry in the reference database, (2) were represented with

at least 10 read counts in a replicate, (3) were present at least 3 times among all sequenced PCR products, (4) showed taxonomic resolution below the phylum level "Bacillariophyta", and (5) were tagged as "internal" by *obiclean* in less than 50 % of all sequenced PCR products to reduce PCR and sequencing artefacts. Filtering with R reduced the number of read counts from 7,536,449 to 6,199,984.

## 2.4 Reproducibility of PCR replicates

Alpha-diversity estimates, such as richness, depend highly on the sequencing depth. As more taxa, especially rare ones, can be detected with increasing sequencing depth, alpha-diversity is only comparable between samples when it is estimated based on the same number of sequences. Despite efforts to reduce such differences by equimolar pooling of PCR products, the number of sequences generally varies among PCR replicates. Therefore, we analyzed the dissimilarities of PCR replicates as well as one sample replicate (8.85 m depth). We resampled the dataset 100 times to the minimum number of sequences

available (25,601 counts), then, for each replicate, we calculated the mean number of sequence counts for each ASV across the 100 resampling steps (code available at: https://github.com/StefanKruse/R_Rarefaction; Kruse, 2019). This dataset was used to calculate the proportions of each ASV per replicate. The proportional data were used for the Multiple Response Permutation Procedure (MRPP) using the R function *mrpp* on Bray Curtis Dissimilarities to test if dissimilarities within replicates of the same sample are significant. Furthermore, we applied non-metric multidimensional scaling (NMDS) using

*metaMDS* to assess which replicates show high and low replicability.

## 2.5 Taxonomic composition and richness

As a measure of alpha-diversity we calculated richness of (1) ASVs (number of amplicon sequence variants) and (2) unique taxonomic names (number of grouped ASVs that were assigned to the same taxonomic name). For taxonomic composition and richness calculations, we combined the PCR replicates of the corresponding sample. This resulted in a new minimum

number of sequence counts (300,415 counts) that was used for resampling the dataset 100 times according to descriptions in the preceding paragraph. The resampled dataset was used to calculate the relative abundances of each ASV per sample. Finally, ASVs, which were assigned to the same taxonomic name but had different similarities to an entry in the reference database, were summed up to one entry using the R *aggregate* function. Stratigraphic diagrams showing temporal changes in taxonomic composition were generated with *strat.plot*. All statistical analyses and visualizations were prepared with R v. 3.5.0 using the

packages "vegan" (Oksanen et al., 2011) and "rioja" (Juggins, 2012). For correlation analysis we interpolated IP$_{25}$ values using the methods described in Reschke et al. (2019). We transformed the IP$_{25}$ data using the function *zoo* from the "zoo" package (Zeileis and Grothendieck, 2005) and used in the function *CorIrregTimser* using the package "corit" (https://github.com/EarthSystemDiagnostics/corit) (Reschke et al., 2019). The correlation between Chaetocerotaceae and Thalassiosiraceae as well as between IP$_{25}$ and all ASVs was tested for significance using the R function *rcorr* (method Pearson) from the package "Hmisc" (Hollander and Wolfe, 1975; Press et al., 1988).

## 3 Marker specificity, taxonomic resolution, and taxonomic coverage

The amplification of the short *rbcL_76* marker permitted the retrieval of diatom DNA from all samples and was highly specific for diatom sequence variants (95.7 %). Only 4.3 % of all assigned ASVs were assigned to other organismal groups: 4 % Bolidophyceae (a closely related sister clade of diatoms with unicellular, siliceous, flagellated algae), 0.2 % Phaeophyceae (brown algae), and 0.1 % Eustigmatophyceae (photosynthetic heterokonts). This exceeds the marker specificity found in previous studies focusing on lake sediment cores (84 % (Stoof-Leichsenring et al., 2012), 88 % (Stoof-Leichsenring et al., 2014), 90 % (Stoof-Leichsenring et al., 2015)) as well as specificity of a previous *in silico* PCR (90.4 % (Dulias et al., 2017)). Possibly owing to a much shorter marker size, the *rbcL_76* marker also surpassed the usually amplified 18S rDNA markers with regard to specificity for diatoms (Coolen et al., 2007; De Schepper et al., 2019; Kirkpatrick et al., 2016).

The initial dataset contained 1,398 ASVs that were systematically filtered to a final diatom dataset containing 360 ASVs (6,199,984 counts) which were further grouped into 75 unique taxonomic names. The majority of diatom sequences are assigned to polar centric Mediophyceae (79.7 %), followed by pennate Bacillariophyceae (14.5 %), while radial centric Conscinodiscophyceae (1.1 %) make up only a small proportion of the dataset. The majority of ASVs are assigned as low as species level (38.6 %) or genus level (25.8 %) (Fig. 2a), yet the taxonomic resolution (i.e. the taxonomic assignment of ASVs) is limited by the incompleteness of the sequence reference database as indicated by the proportion of ASVs (4.8%) for which assignment is restricted to phylum level (Bacillariophyta).

In total, 18 different diatom families are represented in the final dataset which is dominated by ASVs assigned to the families Thalassiosiraceae (35 %), Bacillariaceae (15.8 %), and Chaetoceraceae (9.2 %) (Fig. 2b). Particularly dominant ASVs are assigned to *Thalassiosira* (21.2 %) and *Porosira* (10.9 %) (resolution only possible to genus level) and to *Chaetoceros* cf. *contortus* 1SEH-2013 (6.6 %) – all centric diatoms belong to the class Mediophyceae. Their dominance is likely the result of high paleoproductivity or of differential overrepresentation caused by preservation and/or technical biases. Indeed, in the Fram Strait, *Chaetoceros* and *Thalassiosira* species especially have high productivity in different hydrographical regimes (Gradinger and Baumann, 1991; Lalande et al., 2013) as well as at Kongsfjorden, a major outlet of western Svalbard potentially influencing the coring site (Hodal et al., 2012), and even in micropaleontological studies they are often dominant (Birks and Koç, 2002; Bylinskaya et al., 2016; Oksman et al., 2017, 2019). Next to high paleoproductivity, the preservation of *sed*aDNA could be biased by differential degradation. Heavily silicified and/or spore forming diatoms such as some *Thalassiosira* and

*Chaetoceros* species may be less sensitive to dissolution, which might also improve the preservation of DNA over long time periods. Furthermore, the enrichment of centric diatoms in the *sed*aDNA record could be the result of copy number variation of the *rbcL* gene between different species and cell biovolume (Bedoshvili et al., 2009; Round et al., 1990; Vasselon et al., 2018). Technical biases often arise during PCR due to mismatches between primer sequences and primer binding sites (Nichols et al., 2018) and the high number of PCR cycles – which is needed to increase the chance of retrieving rare sequences – leads to an over-amplification of already abundant template molecules in comparison to rare ones. While the reduction of cycles could reduce this effect, fewer PCR cycles would reduce replicability (Krehenwinkel et al., 2017; Nichols et al., 2018).

Despite the filtering, several distinct ASVs, (e.g. *Chaetoceros socialis*) are assigned the same taxonomic name (Suppl. Table S1, Suppl. Fig. S1). We believe that these represent either different lineages or closely related species so far not included in the database, although we cannot rule out that some ASVs in our filtered dataset might still represent PCR or sequencing artefacts.

Despite the current limitations, *sed*aDNA shows much promise. New reference sequences are added to GenBank on a daily basis, due to numerous phylogenetic studies and barcoding projects aiming to improve our systematic knowledge about taxonomic relationships and to archive the molecular inventory of global biodiversity (Degerlund et al., 2012; Li et al., 2015; Luddington et al., 2016). Taxonomic coverage of the reference database can be increased by subjecting diatom strains to single cell sequencing (Luddington et al., 2016; Sieracki et al., 2019). Most importantly, *sed*aDNA allows tracing ASVs through time and eventually relating them to, for example, environmental change, without relying on the state of taxonomic coverage of the reference database.

## 4 Quality and replicability of the data obtained by *sed*aDNA metabarcoding

A crucial requirement for the interpretation of ancient DNA records is that PCR replicates show similar signals in richness and taxonomic composition. The recovery of taxa by *sed*aDNA metabarcoding is prone to false presences or absences. As our study lacks a morphological diatom record, false absences cannot be assessed and true presences cannot be confirmed, which means that absence in our record does not necessarily translate to a true physical absence in the past. Hence, it was important to use independent PCR replicates for each sediment sample and stringent criteria to filter the dataset to remove artefacts introduced by ancient DNA damage, PCR, and sequencing. The PCR replicates (different PCR products from the same DNA extract) of each sample show some variations (Fig. 3) in the presence and abundance of ASVs, especially for ASVs amounting to less than 1 % per sample (Suppl. Table 2, Suppl. Figure S2). We tested whether these differences are significant using MRPP (p = 0.001, number of permutations 999, observed delta = 0.3683, expected delta = 0.6548), which suggests that PCR replicates of the same sample share significantly lower dissimilarities (average 39.3 %) compared to replicates between different samples (67.5 %). The PCR replicates are highly similar in the oldest samples up to 5.8 m depth and in the youngest sample. For the oldest sample at 8.85 m depth we additionally processed a sample replicate. The PCR replicates of both sediment samples at 8.85 m depth are highly similar and cluster tightly together in the NMDS plot (Fig. 3). Although a higher

number of replicates would improve the robustness of our analysis, this indicates that reasons other than sample age and associated DNA degradation controlled replicability in this study. It is possible that higher dissimilarities between PCR replicates are the result of low amounts of template molecules.

## 5 Temporal change of taxonomic richness and composition

We used *sed*aDNA metabarcoding on samples that, due to the low sample size, represent temporally restricted snapshots of the different climatic intervals since the Late Weichselian. The dataset reveals diatom taxa that have mostly been reported from the Fram Strait in modern surveys (Karpuz and Schrader, 1990; Oksman et al., 2019; von Quillfeldt, 2000) and micropaleontological records (Oksman et al., 2017; Stabell, 1986). We detect typical ice-associated (sympagic) and/or cold-water diatoms (e.g. *Nitzschia* cf. *frigida*, *Cylindrotheca closterium*, *Thalassiosira nordenskioeldii*, *T. gravida*, *T. antarctica*, (Hasle, 1976; von Quillfeldt, 1997; von Quillfeldt et al., 2003), the epiphytic *Attheya septentrionalis* (Poulin et al., 2011) and *Pseudo-nitzschia granii* (Lovejoy et al., 2002)) alongside the temperate to warm-water species *Detonula pumila* (Hasle, 1976) and *Thalassiosira angulata* (Krawczyk et al., 2013; Luddington et al., 2016; Weckström et al., 2014), and some cosmopolitans (*Minidiscus trioculatus, Cerataulina pelagica* (Hasle, 1976)). Beyond marine diatoms, sequences are also assigned to species preferring fresh to brackish water (e.g. *Skeletonema subsalsum* (Hasle and Evensen, 1975), *Nitzschia palea, N.* cf. *paleaceae* (Husted, 1930)). Taxonomic composition and richness changes with core depth and fits well into the framework reconstructed by other proxy data (biomarkers (Müller et al., 2012; Müller and Stein, 2014); foraminifers (Werner et al., 2011, 2013, 2016; Zamelczyk et al., 2014); dinoflagellates (Falardeau et al., 2018)).

Generally, the richness of both ASVs and unique taxonomic names (ASVs grouped based on identically assigned taxonomic names) is higher in samples dated to the last glacial in comparison to those dated to the Holocene (Fig 4). A shift of diatom *sed*aDNA composition  is captured with some ASVs being predominantly abundant in Late Glacial samples (e.g. those assigned to *Thalassiosira gravida*, *Minidiscus trioculatus,* and *Nitzschia* cf. *paleacea*) whereas others are mainly present in Holocene samples (e.g. *Chaetoceros* cf. *contortus* 1 SHE-2013, *C.* cf. *pseudobrevis* 1 SHE-2013) (Fig. 5). This shift is also strongly reflected at the family level (Fig. 6). A general trend that can also be observed at the family level is an inverse relationship (r = -0.61, p = 0.046) of the dominant families Thalassiosiraceae and Chaetoceraceae (Fig 6).

Samples dated to the Late Weichselian and the last glacial maximum (LGM) are characterized by highest overall richness with regard to ASVs (Fig.4). The samples contain high proportions of sympagic taxa (*Thalassiosira gravida* (4.8–28.1 %), *T. antarctica* (4.9–17.9 %), *T. delicata* (3.1–7.5 %), *Chaetoceros socialis* (0.3–7 %), *Nitzschia* cf. *frigida* (0.9–3.3%), *Porosira* (6–10.4 %)) as well as littoral (*Haslea avium* (1.3–3.4 %)), oceanic  (*Minidiscus trioculatus* (1.4–4.4 %)), and brackish to freshwater taxa (*Nitzschia* cf. *paleacea* (0.3–5.9%), *Skeletonema subsalsum* (1–2.6 %)) (Fig. 5). Overall, ASVs assigned to the families Thalassiosiraceae, Bacillariaceae, and Naviculaceae (Fig. 6) dominate these samples. The relatively high proportions of sympagic taxa, especially of *Nitzschia* cf. *frigida*, in samples dated to the Late Weichselian and LGM  are in accordance with previously reconstructed cold sea-surface conditions based on high proportions of the polar planktic

foraminifer *Neogloboquadrina pachyderma* (Zamelczyk et al., 2014), low dinocyst concentrations with a dominance of phototrophic taxa (Falardeau et al., 2018), and moderate concentrations of both the sea-ice proxy $IP_{25}$ and the phytoplankton biomarker brassicasterol (Müller and Stein, 2014). It is conceivable that a heterogeneous and dynamic environment produced by winter sea-ice cover with ice-free conditions during summer could allow diverse diatom communities to develop in the

260 different habitats and over the seasons, which is suggested by the highest overall numbers of ASVs in samples dated to this time span. *Thalassiosira* and *Chaetoceros* are also abundant in the LGM microfossil assemblage identified further south at the Knipovich Ridge (eastern Fram Strait), although species of the genera *Rhizosolenia* and *Coscinodiscus* are strongly represented there as well (Bylinskaya et al., 2016).

Samples taken from Heinrich Stadial 1 and Bølling/Allerød phases are characterized by high proportions of ASVs assigned to

265 *Thalassiosira* (35.1–65.3 %), *T. antarctica* (6.8–9.8 %), and *T. delicata* (4.1–5.3 %), and abundant *Skeletonema subsalsum* (2.1–5.6 %), whereas proportions of *Thalassiosira gravida* (3.4–11.3 %), *Nitzschia* cf. *frigida* (0.6–0.7 %), *Haslea avium* (0.2–0.4 %), and *Minidiscus trioculatus* (0–0.7%) are lower compared to the LGM. The taxonomic composition of the sample dated to Heinrich Stadial 1 suggests a partial re-organization of the diatom *sed*aDNA composition, which took place either gradually or abruptly sometime between 20.5 and 15.1 cal kyr BP. The low proportions of *Nitzschia* cf. *frigida* and other cold-

270 water and sea-ice diatoms in samples corresponding to Heinrich Stadial 1 and Bølling/Allerød correspond well to low $IP_{25}$ and moderate brassicasterol concentrations in these samples, reflecting a reduction of the sea-ice cover (Müller and Stein, 2014). In the Bølling/Allerød sample the relatively high abundance of sequences assigned to the brackish–freshwater preferring diatom *Skeletonema subsalsum* (Hasle and Evensen, 1975) might be explained by elevated meltwater discharge from Svalbard. Low surface-water salinity was inferred previously from the dinocyst record of this core about a century earlier, possibly

resulting from melting of the Barents Sea ice-sheet (Falardeau et al., 2018). Higher sedimentation rates and thus higher temporal resolution during the Bølling/Allerød could have affected the *sed*aDNA signal, yet the distinct shift in taxonomic composition suggests that this is rather an effect of the changing environmental conditions during this phase.

The Younger Dryas sample exhibits moderate proportions of *Thalassiosira antarctica* (11.8 %) and *T. delicata* (2.3 %) and is marked by the presence of the sympagic diatom *Cylindrotheca closterium* (0.3 %) and relative increases in *Porosira* (9.1 %),

*Haslea avium* (5.7 %), and *Nitzschia* cf. *frigida* (1.3 %). Higher proportions of sympagic diatoms in this sample point towards colder conditions in comparison to the Bølling/Allerød sample and the presence of sea ice. Severe and extended sea-ice cover at the coring site are indicated by heterotrophic species in the dinocyst record (Falardeau et al., 2018), peak concentrations of $IP_{25}$, and very low brassicasterol concentrations (Müller and Stein, 2014). A diatom microfossil record from Hinlopen Strait northwest of Spitzbergen detects first diatom occurrence during the Younger Dryas (10.8 [14]C kyr BP) with more than 30 %

sea-ice associated species yet with a different taxonomic composition in comparison to our record (Koç et al., 2002).

The Early Holocene sample, which is marked by peak proportions of several families such as Skeletonemaceae (19.6 %) and Bacillariaceae (18.3%) and considerable increases of Chaetocerotaceae from 6.5 to 32.4 % combined with a strong decrease of Thalassiosiraceae from 57.8 to 44.4 % and Naviculaceae from 5.7 to 0.2 %, points towards a second partial re-organization of the taxonomic composition between 12.8 and 10.1 cal kyr BP. In particular, the sample is dominated by sequences assigned

to *Skeletonema* (16.8 %), *Thalassiosira* (14.5 %), and *Pseudo-nitzschia granii* (11.4 %). *Thalassiosira angulata*, a species associated with low sea-ice concentrations (Oksman et al., 2019) and temperate water masses (Weckström et al., 2014), displays peak proportions (1.7 %) in this sample. Lower proportions of sympagic diatoms in this sample are in accordance with the sea-ice retreat reconstructed from low $IP_{25}$ and high brassicasterol concentrations (Müller and Stein, 2014) and high proportions of the subpolar planktic foraminifers *Turborotalita quinqueloba* in another core further northwest (Werner et al., 2016). The diatom composition recorded by *sed*aDNA is also quite different to what has been found in microfossil records from the Fram Strait and the Greenland, Iceland, and Norwegian Seas, where first diatom microfossils are recorded between 13.4 and 9 cal kyr BP (Koç et al., 1993; Schröder-Ritzrau et al., 2001). Southwest of Svalbard, diatom-rich sediments dated to 10.1 and 9.8 cal kyr BP are attributed to the inflow of Atlantic surface water and a retreat of the Polar Front (Jessen et al., 2010; Stabell, 1986) and are composed mostly of *Coscinodiscus* spp., *Rhizosolenia,* and *Paralia sulcata* (Jessen et al., 2010; Rigual-Hernández et al., 2017; Stabell, 1986). In concordance with this, sequences assigned to *Paralia* have highest proportions in the Early Holocene sample. *Coscinodiscus* is likely resolved to class level as sequences (Coscinodiscophyceae), and thus are present in our record albeit only sparsely. *Rhizosolenia* was not detected in this sample, which could be explained by poor DNA preservation or regional differences of the past diatom communities between the two coring sites.

The sample dated to the Mid-Holocene is marked by low diatom richness and by peak proportions of the families Chaetoceraceae (32.4 %) and Anomoeneidaceae (2.1 %). It is dominated by sequences assigned to *Chaetoceros* cf. *contortus* 1SEH-2013 (27 %), *Thalassiosira* (23.3 %), and *T. antarctica* (9 %), accompanied by abundant *Porosira* (7.9 %) and *Skeletonema* (3.9 %). The relatively low richness suggests a loss of diversity between the Early and Mid-Holocene. The Mid-Holocene sample is characterized by a diatom composition that cannot be clearly related to sea ice, yet the low richness is supported by low diatom concentrations in a sediment core from Mohn Ridge (Koç et al., 1993) and low phytoplankton productivity (Müller et al., 2012). The near absence of *Nitzschia* cf. *frigida*, however, does not match with the reconstructed strong sea-surface cooling and sea-ice growth from lower proportions of *Turborotalita quinqueloba* (Werner et al., 2011, 2013), low concentrations of $CaCO_3$, and high concentrations of $IP_{25}$ and ice-rafted detritus (Müller et al., 2012).

The samples dated to the Roman Warm Period and the Little Ice Age (Late Holocene) have a similar richness to that of the Mid-Holocene sample. This phase is characterized by cold-water and ice-associated taxa with peak proportions of *Porosira* (15.1–22.9 %) accompanied by significant proportions of *Chaetoceros* cf. *contortus* 1SEH-2013 (3.6–14.9 %), *Attheya* (2.3–6.7 %), *Thalassiosira antarctica* (1.4–4.7%), *T. delicata* (4.2–6.3 %), and *Chaetoceros socialis* (0.9–3.6 %). Richness is lowest in the youngest investigated sample which was dated to the Little Ice Age. The sample is characterized by a peak of *Chaetoceros* cf. *pseudobrevis* 1SEH-2013 (3.9 %), *Attheya* (6.7 %), and the sympagic diatoms *Porosira* (15.1 %), *Thalassiosira nordenskioeldii* (1.1 %), and *Nitzschia* cf. *frigida* (5.3 %). The Late-Holocene samples are characterized by elevated proportions of sympagic taxa in comparison to the Mid-Holocene sample which is in agreement with Neoglacial cooling (increasing $IP_{25}$ concentrations and moderate to high concentrations of brassicasterol (Müller et al., 2012), increases in ice-rafted detritus and the dominance of polar planktic foraminifers (Werner et al., 2011)). Diatom microfossil records located southwest of Svalbard display an increase in diatom abundance since approximately 1.5 cal kyr BP (Rigual-Hernández

et al., 2017; Stabell, 1986). The microfossil record of Rigual-Hernández et al. (2017) is mostly composed of *Chaetoceros*
resting spores which matches the increase of Chaetocerotaceae in our *sed*aDNA data. The composition of the record published
by Stabell (1986) is more similar to Rigual-Hernández et al.'s (2017) Early-Holocene diatom maximum with *Coscinodiscus*
spp., *Rhizosolenia,* and *Paralia sulcata*. Differences between composition and diversity of our samples and the records of
Rigual-Hernández et al. (2017) and Stabell (1986) suggest either regional differences or differential preservation of
microfossils and *sed*aDNA between the sites.

## 6 Potential of diatom *sed*aDNA as a proxy for sea-ice distribution

The *sed*aDNA record generally contains a high proportion of sequences assigned to cold-water and sea-ice associated diatoms,
such as *Nitzschia* cf. *frigida*, *Thalassiosira antarctica*, and *T. nordenskioeldii* (Hasle, 1976; Poulin et al., 2011; von Quillfeldt,
1997; von Quillfeldt et al., 2003). Furthermore, pennate diatoms, which often dominate bottom ice layers (Van Leeuwe et al.,
2018), display higher proportions in samples dated to the LGM, the Younger Dryas, and the Late Holocene with moderate $IP_{25}$
concentrations (Fig. 6). Among ASVs, *Nitzschia* cf. *frigida* (ASV 709), *Attheya* (ASV 28), and Bacillariophyta (ASV 154)
show increased proportions with $IP_{25}$ exceeding 0.8 µg g$^{-1}$ organic carbon (Suppl. Fig. S3). In contrast, Bacillariophyta (ASV
245), *Gomphonema* (ASV 586), and *Thalassiosira* (ASV 1017) are detected only in samples with $IP_{25}$ of less than 0.5 µg g$^{-1}$
sediment.

The sea-ice proxy $IP_{25}$ is produced by only a few known ice-associated diatoms: *Haslea kjellmanii* (Cleve) Simonsen, *H.*
*spicula* (Hickie) Lange-Bertalot, and *Pleurosigma stuxbergii* var. *rhomboides* (Cleve in Cleve and Grunow) Cleve (Brown et
al., 2014; Limoges et al., 2018). We detected none of the known producers due to the incompleteness of the reference database
(for the species of *Haslea*) and/or because the DNA of these species was not preserved in sufficient quantities as these species
are low in abundance (Brown et al., 2014). Only for *Pleurosigma stuxbergii* is there a publicly available reference containing
our marker region and is included in our database, whereas 15 references for non-$IP_{25}$ producing *Haslea* species are available
in our database. Hence, an absence in our record does not mean an absence in the past communities. Sequences assigned to
*Haslea avium* in our sedaDNA record do not show a linear relationship with $IP_{25}$, but are nevertheless present in most samples.
Our data, in combination with other proxy data from this core, suggest that sequences assigned to *Nitzschia* cf. *frigida* could
be a useful indicator of past sea-ice distribution in *sed*aDNA records. Samples with low concentrations of the sea-ice proxy
$IP_{25}$ (< 0.4 µg g$^{-1}$ total organic carbon) have less than 2 % of sequences assigned to *Nitzschia* cf. *frigida*, whereas samples
characterized by moderate to high (> 0.8 µg g$^{-1}$ total organic carbon) concentrations of $IP_{25}$ display highly variable proportions
without a clear relationship. *Nitzschia frigida* (Grunow) forms arborescent colonies (Medlin and Hasle, 1990) and is often
abundant from late winter in the bottom layer of nearshore first-year ice and in multi-year ice in the Arctic pack-ice zones
(Krawczyk et al., 2017; Melnikov et al., 2002; Olsen et al., 2017; Poulin et al., 2011; von Quillfeldt et al., 2003), but it can
also be found in the water column during vernal under-ice or ice-edge blooms (Hasle and Heimdal, 1998; Olsen et al., 2017).
Recently, De Schepper et al. (2019) used *sed*aDNA metabarcoding on a core from the East Greenland Sea and identified the

Mediophyceae OTU_5051 which was significantly correlated to the $IP_{25}$ concentrations. The class Mediophyceae is the most diverse and dominant group in our dataset and contains several sea-ice associated taxa, such as *Thalassiosira antarctica*, *T. nordenskioeldii*, and *Porosira glacialis*. Yet their tendency to prevail in both sea ice and open water might be responsible for the non-linear relationship with $IP_{25}$ observed in our record. This is supported by Weckström et al. (2013) who find no specific response of sea-ice diatom microfossil composition to either $IP_{25}$ concentrations or observational sea-ice data in the Labrador Sea.

## 7 Conclusions

For the first time in a marine environment, our study targets high-resolution, diatom-specific sedimentary ancient DNA using a DNA metabarcoding approach. We show that diatom DNA is preserved with substantial taxonomic richness in the eastern Fram Strait over the past 30,000 years even though diatom microfossils are recorded in the Svalbard region only since the Younger Dryas. This highlights the advantage of our approach for paleoenvironmental reconstructions aiming to identify drivers of community-level taxonomic composition and diversity, especially in regions known for their irregular and poor diatom microfossil preservation, such as the Fram Strait. The *rbcL_76* marker is highly diatom specific and provides detailed taxonomic resolution, mostly at genus and species level. The shortness of this marker is a strong advantage that leads to adequate replicability and high quality as diversity patterns do not show conspicuous signs of bias by age-associated DNA fragmentation. The *sed*aDNA record captures substantial temporal change of diatom taxonomic composition and richness with four compositional re-organizations: the first between 20.51 and 15.1 cal kyr BP, the second between 12.8 and 10.1 cal kyr BP, the third between 10.1 and 5 cal kyr BP, and the fourth between 5 and 1.6 cal kyr BP. Our record extends diatom compositional and diversity information back to the Late Weichselian as microfossil records in the Fram Strait are rare and extend only as far as the Younger Dryas due to poor preservation. Increasing proportions of pennate diatoms are associated with increased $IP_{25}$ concentrations, and sympagic diatoms are present, but with no clear pattern with regard to biomarker signals. Recommendations for future work with sedimentary ancient DNA in the context of sea-ice reconstructions include the preparation of reference genomes and a more targeted enrichment, for example of genes that help species to adapt to sea ice and allow them to cope with rapidly changing environmental conditions.

## Supplement

### Code/Data availability

The sequence data are deposited in the Sequence Read Archive (BioProject XXX). The rarefaction script is available at https://github.com/StefanKruse/R_Rarefaction (Kruse, 2019).

**Author contributions**

H.H.Z., K.R.S.L., and U.H. conceived and designed the study; H.H.Z. performed experiments and data analysis; K.S.L. sampled the core and provided laboratory equipment; S.K. wrote resampling R script; R.S., R.T., J.M. and U.H. provided the framework for the study (samples, funding); H.H.Z. wrote the paper that all co-authors commented on.

**Competing interests**

The authors declare that they have no conflict of interest.

**Acknowledgements**

We thank Sarah Olischläger and Iris Eder for support with the laboratory work and the captain and crew of the RV Maria S. Merian. This research was funded by the Initiative and Networking Fund of the Helmholtz Association. U.H. is financed by the European Research Council (ERC) under the European Union's Horizon 2020 research and innovation programme (grant agreement No 772852, GlacialLegacy Project). J.M. received financial support from a Helmholtz Research Grant VH-NG-395 1101. We thank Cathy Jenks for English correction.

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

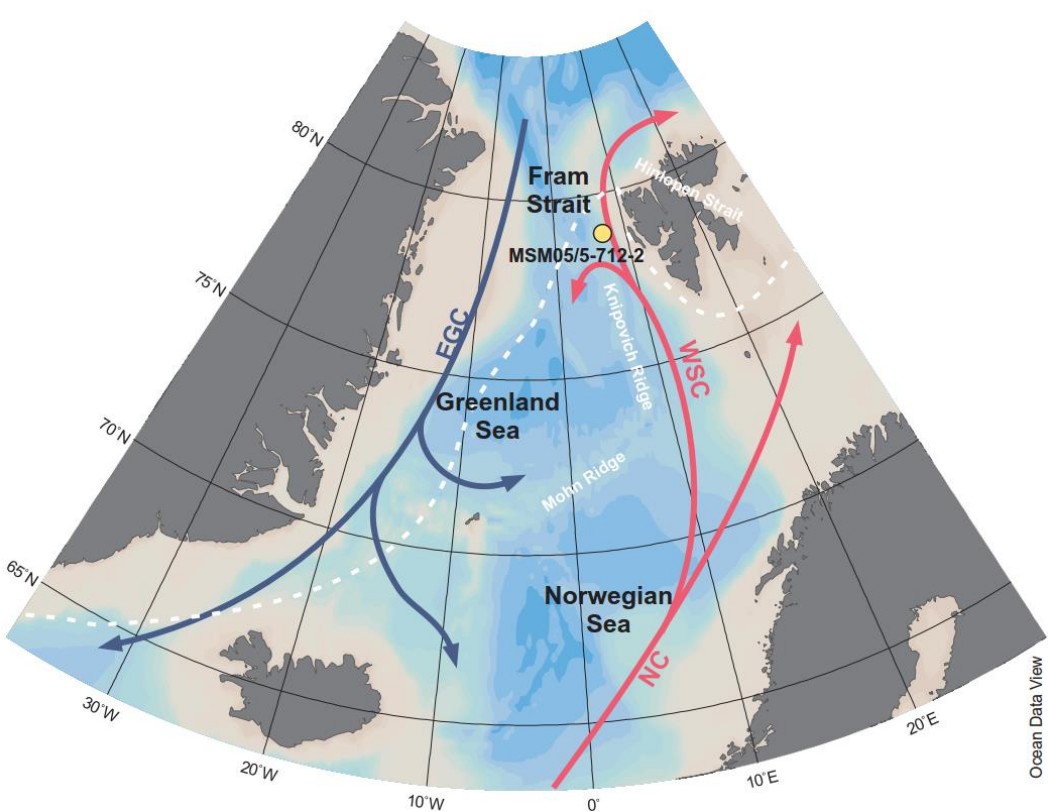

Figure 1: Map showing the coring site of MSM05/5-712-2 with bathymetric data derived from Ocean Data View (Schlitzer, 2002) and median March sea-ice extent from 1981-2010 (white, dashed line; https://nsidc.org/data/seaice_index/archives [accessed 06.08.2019] (Fetterer et al., 2017)).

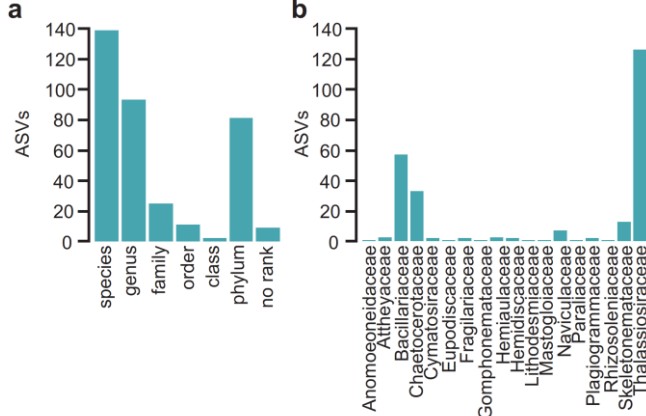

Figure 2: Number of diatom amplicon sequence variants (ASVs) assigned (a) to different taxonomic levels and (b) to different families.

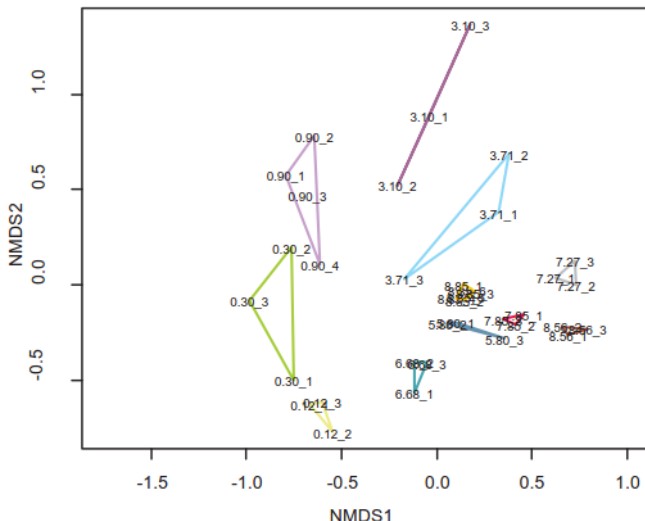

**Figure 3: Non-metric multidimensional scaling plot based on the filtered and resampled diatom dataset with the PCR replicates (indicated by an underscore with a number) of a sample (depth in m) linked in a polygon of sample-specific colour: light green = 0.12m, green = 0.3m, light violet = 0.9 m, violet = 3.1 m, light blue = 3.71 m, dark blue = 5.8 m, turquoise = 6.68 m, grey =7.27 m, red = 7. 85 m, brown = 8.56 m, orange = 8.85 m (2 sample replicates with 3 PCR-replicates each).**

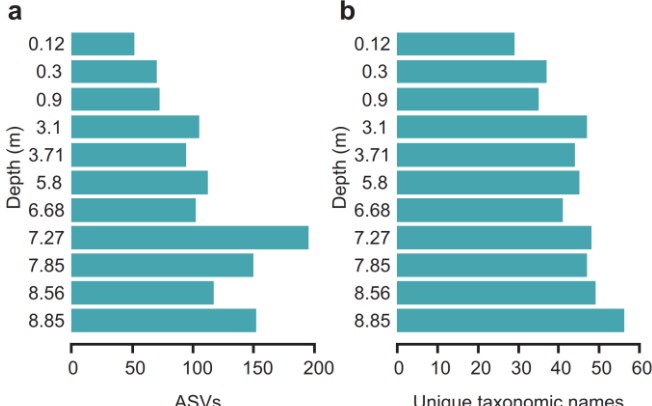

**Figure 4: Barplots showing the rarefied (a) number of amplicon sequence variants (ASVs) per sample and (b) grouped ASVs assigned to the same taxonomic name for each sample with depth (m) of the sediment core MSM05/5-712-2.**

**a**

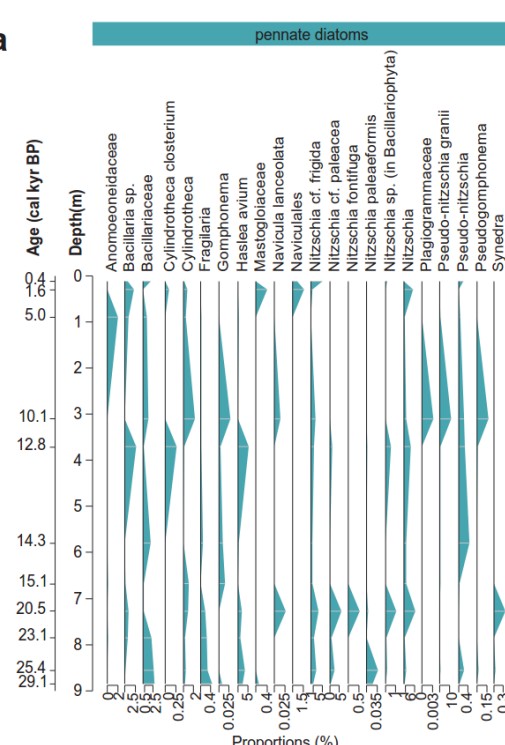

pennate diatoms

**b**

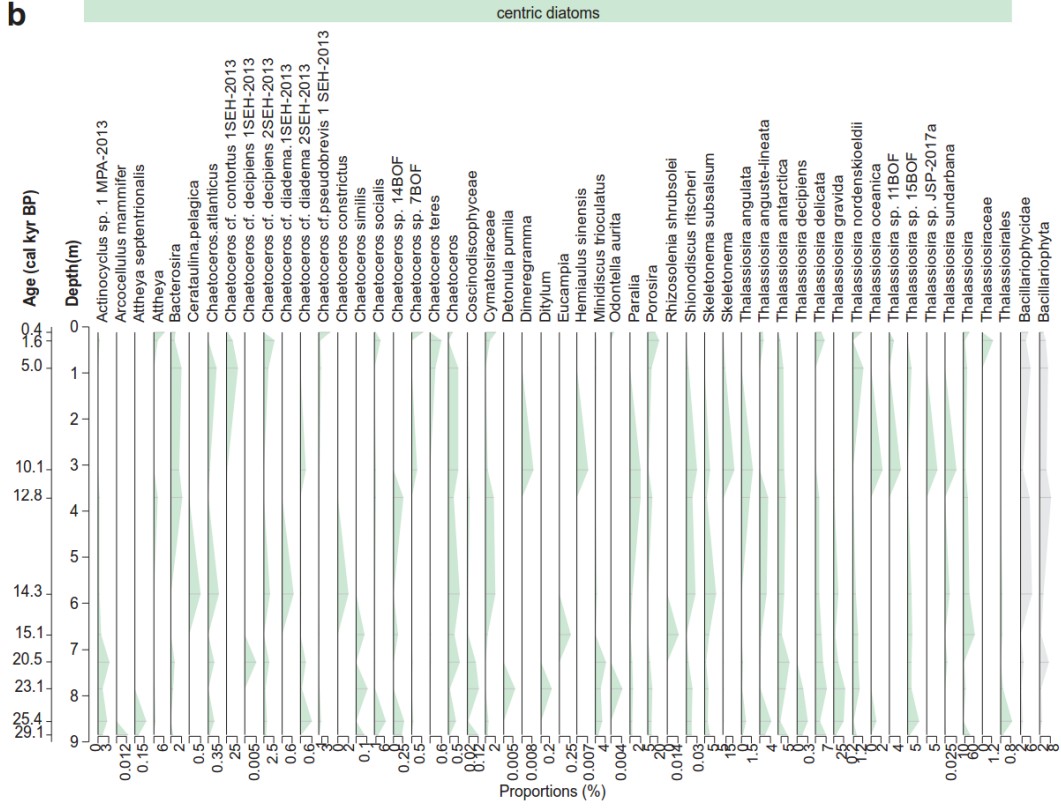

centric diatoms

Figure 5: Stratigraphic diagrams with sequences assigned to (a) pennate diatoms (blue) and (b) centric diatoms (green) and higher level sequences assigned between family and phylum level (gray) The taxonomic composition with relative proportions (%) of the 360 detected sequence variants is grouped into 75 unique taxonomic names based on identically assigned taxonomic names of sediment core MSM05/5-712-2.

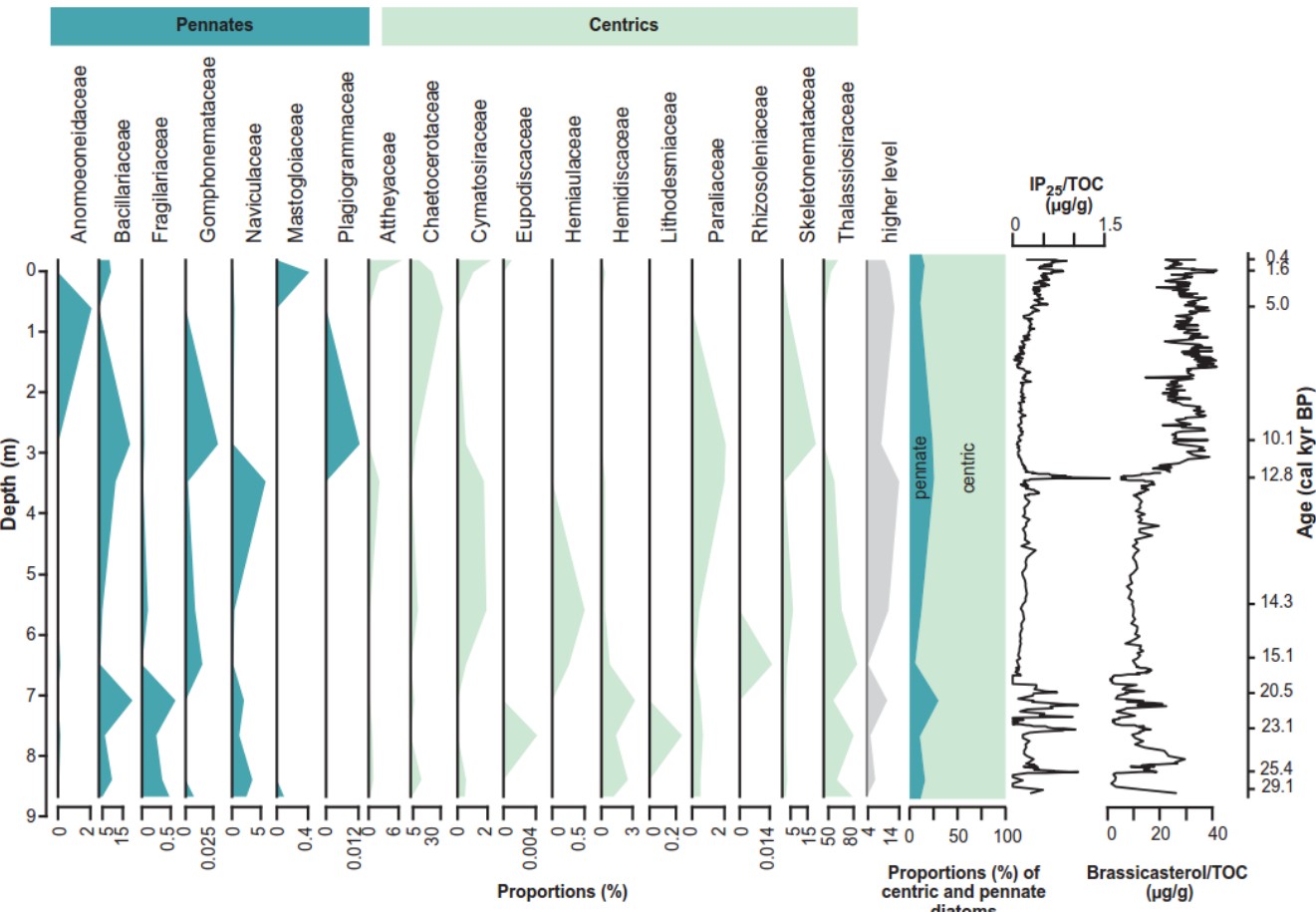

Figure 6: Proportions of sequences assigned to diatoms grouped on family level and down-core proportions of centric (blue) and pennate diatoms (green) as well as concentrations of the sea-ice biomarker $IP_{25}$ (Müller et al., 2012; Müller and Stein, 2014) and the phytoplankton biomarker brassicasterol (Müller et al., 2012; Müller and Stein, 2014). Higher level (gray) contains sequences assigned between family and phylum level. TOC = total organic carbon.