# Peer review of "Changes in the composition of marine and sea-ice diatoms derived from sedimentary ancient DNA of the eastern Fram Strait over the past 30,000 years"

_Ocean Science, 2019_

## Referee Comment (RC1) · Jessica Louise Ray (Referee) · 3 Jan 2020

Note to all: This is my first foray into the open review process, and my first non-anonymous review. I hope that my review meets expectations.

Summary This paper expounds on recent advances in the use of marine sedimentary ancient DNA (sedaDNA( as a proxy for sea ice reconstructions by investigating diatom assemblages in an Eastern Fram Strait downcore extending back to the Late Weichselian period (approx 30 kya). The authors contribute this novel diatom sedaDNA rela-

tive abundance data to existing biomarker (IP25 and brassicasterol) proxy data from the same core. The three research objectives of the study are to assess diatom sedaDNA quality and reproducibility, to assess diatom assemblage dynamics over time, and to explore diatom sedaDNA as a new sea-ice proxy.

General comments I would first like to congratulate the authors on an extremely well-written manuscript. The structure is both logical and well-rounded, the language is accessible yet precise, the literature cited is appropriate and well-updated, and the inclusion of the comprehensive dataset (Table S1) bespeaks a desire to promote science through the sharing of raw data.

Specific questions/comments l.127 - Less than 50% of 3 replicates?

l.147 - What was the constraining factor used for CONISS analyis? Please include.

l.201 - What are the rare ASVs? A quick perusal of the Table S1 indicates many zeros. Would it be possible or useful to include a supplementary figure showing rank-abundance curves, either for the entire dataset or for the individual samples?

l.205 - Interesting theory, but why would inhibition only appear in some samples? I think this might be easy to test by a dilution PCR of a few representative samples. Also, if there are differences in the relative amount of diatom DNA in each sedaDNA sample, would it be possible to do a quick qPCR check of rbcL target abundance using the same primer set?

l.209 (Section 5) - My first impression from reading the results is that the authors have some difficulty in interpreting diatom community assemblage differences according to the results of the CONISS analysis, i.e. division into five aggregate "zones". In particular because trends in the relative abundance of specific sympagic diatom taxa between different CONISS "zones" are mentioned but not statistically tested. This leads me to question whether CONISS analysis is useful given the present dataset. Were any other analyses attempted to identify discriminant ASVs/taxa? And how did the authors conclude that there are two diatom assemblage reorganizations (l.340) when the CONISS analysis identifies four?

l.223 - Could you please elaborate on what is meant by "richness of taxonomic names". According to l.184 different ASVs can be assigned to the same name, so how might this affect the apparent richness? And why use "taxonomic names" for richness calculations when taxonomic rank assignment is not uniform across all ASVs?

l.224 - I am not quite comfortable with the use of "turnover" in the context it is used, since the samples are discrete and therefore a discontinuous representation of time. In my opinion, "turnover" suggests a biological/ecological linkage from one sample to the next, while in this study the samples compared are isolated snapshots in time. Can the authors comment on the choice to use this term?

Fig.5 - I wonder if the reader might find this figure somewhat difficult to interpret given that relative taxon abundance at multiple taxonomic ranks are presented for each sample. Could the authors please the reasoning for presenting the data in this way?

Table 1 - The Paleoenvironmental conditions descriptions seem somewhat arbitrary. For example, how is "sea-ice retreat" (3.1 m depth) different from "Reduced sea-ice cover allowing spring sea-ice algal and summer phytoplankton productivity" (7.85 m depth).

l.239 - foraminifer

Fig.6 - Family-level taxonomy is presented but l.170-174 states that this collation may mask functional differences.

Fig.6 - The double top axes (depth and age) are very helpful, but identify very clear differences in sedimentation rates in the downcore. For example, zone II has higher temporal resolution than zone I. Might not differences in sedimentation rates also affect sedaDNA signal?

l.247 - Again, ambiguous results from CONISS analysis?

l.336 - "highly detailed taxonomic resolution" depends on what is meant by highly detailed, and what fraction of the data is being referred to. I suggest moderating this statement.

l.320 - Very interesting that N. cf. frigida sticks out as a possible new sea-ice proxy. However, according to Fig.6, this taxon as highest relative abundance in the most recent sample (0.4 kya BP) when sea-ice cover is low. What about Cylindrotheca closterium? Or Haslea avium? Again, I think it would be very helpful if taxon relative abundances were statistically tested in order to identify ASVs/taxa that contribute significantly the observed diatom diversity in different samples.

---

## Short Comment (SC1) · 26 Feb 2020

In this study, Zimmermann et al. present DNA profiles associated with marine diatoms from selected samples of a well-dated and well-investigated sediment core. This core was previously investigated for the sea ice biomarker IP25, which allows the authors a comparison of their sedaDNA record with this well established sea ice proxy.

The manuscript is well written and will contribute to the understanding for sedaDNA for paleoenvironmental reconstructions.

[Figure]

I have only few comments on the manuscripts, and I want to point out that I cannot evaluate the quality of the DNA analysis as I my expertise is with other methods.

Comments:

The term "richness" is used very often, and I have difficulties understanding that term. Could you please add some information on this term, as you use it very often.

Abstract: The Abstract is missing some information on the investigated material/samples.

Introduction: I have the feeling other sea ice proxies should be mentioned. Further, some information on the advantages of the sedaDNA study you present here is missing. It is not clear why sedaDNA may be of advantage for sea ice reconstructions, as the sea ice biomarker IP25 is not as prone to dissolution as microfossils.

Chapter 6: I feel this chapter needs more detail. The mismatch of IP25 and sedaDNA was to be expected, as you do not investigate the IP25 producers. Could you elaborate more on the reasons for the mismatch of biomarker and sedaDNA record? What about seasonality of N. frigida and IP25 production? From SW Greenland, Krawzcyk et al (2015; Polar Biology) found that N. frigida is most abundant in late winter before the main spring bloom – which is expected to be the main production season of IP25 in Fram Strait. What about habitats (under the ice, inside the ice) between the different species? And finally what are your recommendations for future work or when using sedaDN for sea-ice reconstructions?

L36 Krawczyk et al., 2017 should be added here

L59 & L60 overuse of whether

L82 I find the description of the sea ice condition in the working area confusing.

L82 mentioning past sea ice variability feels wrong here, maybe add this information with some more detail to the introduction

L86 should say Epp et al. (2019)

L116 should say Callahan et al. (2017)

L120 should say Dulias et al. (2017) and Stoof-Leichsenring et al. (2012)

L134 I do not understand this Quote.

L150-159 I feel the information on lake studies has too much detail whereas the information on marine studies is too short as the presented study is marine.

L198 This is a major problem. However cannot be changed for your study but I welcome this comment. For future studies the parallel investigation of biomarkers and diatoms in the microfossil and genetic record may be a very promising approach.

I hope these comments help to improve the already good manuscript. All the best, Henriette Kolling

———————————————————

---

## Referee Comment (RC2) · John B. Kirkpatrick (Referee) · 15 Apr 2020

General comments: Zimmerman et al. present a novel dataset that relates previously published information on climate records from the waters near Svalbard to newly obtained diatom chloroplast DNA sequences found in the sediment dating up to 30,000 years ago. The attempt not just to detect and sequence old DNA but relate it to known proxies and paleoenvironmental conditions is an exciting step toward realizing the promise these techniques have been hoped to provide. That said, I have some questions about the samples and methods, and the choice of primer sequences limits

the utility of data to make inferences about the paleo community structure and connect to the IP25 proxy.

Specific comments:

Methods: Section 2.1: The coring equipment and handling should be described. Was this a piston core? What diameter? How was it handled? Were the cores halved onboard? Did the authors use an archive half or a working half that had previously been used for other sampling activities? Most importantly, how was the core stored – at what temperature? If at -80 C, this should be clearly stated. If not at -80 C, it should be acknowledged that DNA preservation ex situ may be imperfect.

Do the authors have any tracers or controls for post-coring seawater influence on the sediment core? I am willing to believe that the authors, working in a dedicated lab, were meticulous about their sampling and extraction. Did they use tracer DNA as described by Epp et al.?

Section 2.2: This is the most significant concern I have about this dataset. The primer set used was designed by Dulias et al. for freshwater lakes. Its specificity for diatoms appears to be very high, as the authors highlight, which is good to address. However, it contains multiple mismatches on both the forward and reverse primers for the taxa noted to contribute to the IP25 proxy (Haslea, Pleurosigma; please see attached "supplement"). (I did not look at other marine taxa, but that should be investigated.) As a result this dataset should not be considered to be a well-rounded assessment of diatoms in this setting. Any of the increases / decreases in richness, or the absence of certain taxa, or overall assessment of diatom diversity are not valid as we are effectively blinded to many of the taxa known to be present in this location (and presumably throughout the sediment record). This could perhaps be partially addressed by compiling an alignment of diatom chloroplast rbcL sequences documented to be or have been present at this location and noting how many taxa you would expect to find do or don't have mismatches with the primers. Maybe the authors are aware of this (?). At the end

of the day, the detection of the sequences found is real data, and valuable. However, a broad assessment of shifts in diversity and community responses to climate is very problematic.

On a different note, the cycle numbers and annealing temperatures, in my opinion, should be included even if they are included in the given citation. 50 cycles is a very large number. Did the authors ever visualize bands in their extraction or PCR negative controls?

Line 110 – what version (chemistry)?

Section 2.3: I'm OK with the use of ASVs and I appreciate the authors taking time to explain why they didn't use OTUs. Regarding their reference database, how many references did this method produce? Did these references include diatoms documented in other studies from the waters around Svalbard? The reference database is a frequent scapegoat in the discussion; release 138 is a couple years ago. The taxonomy seems pretty robust to me but it could be updated if need be.

Going back to the negative controls, on line 122 how many exactly is "the majority" that were singletons? On line 125, why 10 read counts? How many sequences, and how many ASVs, were removed out of the total using these criteria? As the supplemental table only has the kept sequences, one can't tell.

Section 2.4 (and throughout): Please make sure to always use the term "PCR replicate" or "PCR replication", as these are not sample replicates. PCR replication is useful but the terminology should be clear as later on it may get confusing for readers who aren't methods geeks and think that these are sample or extraction replicates.

Section 2.5: Resampling 100 times might be overkill but I guess it can't hurt. The new minimum number of sequence counts doesn't quite make sense to me through – can you clarify "according to descriptions in the preceding paragraph"? (It's not quite 12 * 25,601.)

Section 3: Line 157 – What do you mean by "outperformed"? Are you referring to the specificity for diatoms? Please clarify. Line 164 – "striking proportion" – I don't think it's that striking, in fact, 4.8% seems pretty low! Line 177 – can you give a numeric range for the copy number variation in published work?

Section 4: Here I think again it is important to be clear that the replicates being discussed are PCR replicates of the same template DNA. Also, what happened to the two sample replicates from the deepest sample? How similar or dissimilar were they and does the sample variation exceed the variation found in PCR replication? While two duplicates aren't statistically as useful as a triplicate, they need to be discussed in the context of reproducibility. Doing so would, I think, strengthen the authors' arguments. This should be addressed in revision (unless I missed it somewhere?).

Section 5: Lines 228-299: This inverse relationship, if significant, is intriguing! Please test statistically for significance. Lines 242-244... etc: This sort of speculation about how ice conditions may have affected diatom community diversity, etc., is undermined by the primer issue. Lines 253 and elsewhere: "Low proportions of Nitzchia cf. frigida..." and similar statements should be reworded, as we do not know the proportion of diatoms. We know the proportions of the sequences found in the dataset after extensive amplification (50 cycles!). Please be careful to keep this distinction clear. Line 265: How far away is that (Hinlopen Strait)? In this section in general, there are numerous and interesting comparisons to published records from related locations. It would be appreciated if the proximity of these (in km) were clear for the less-familiar reader. (E.g. also line 276, 278, 291, 304, 323) Line 274: "peak proportions", "Lower proportions" – can you please quantify. (Also e.g. line 301.) Line 289: Richness and diversity are not the same; please be specific.

Section 6: Beyond the methods concerns, there are still some interesting points here. These could be strengthened by a statistical analysis testing the correlation between the new data to the published IP25 proxy (lines 319 - 323).

Technical corrections: Figure 3: Labels are not readable. Figure 5: This appears upside-down. More importantly, the labels and proportion bars are very, very small; but much of the graph is empty whitespace. Please try making the bars wider. Also, consider organizing the taxa by e.g. rank order (based on proportion) and breaking into 2 or 3 separate figures. Figure 56: Please define "higher level" in the caption.

Please also note the supplement to this comment:
https://www.ocean-sci-discuss.net/os-2019-113/os-2019-113-RC2-supplement.pdf
* * *

---

## Author Comment (AC3) · 23 May 2020

**Answers to os-2019-113_SC1**

**Changes in the composition of marine and sea-ice diatoms derived from sedimentary ancient DNA of the eastern Fram Strait over the past 30,000 years**

By Heike H. Zimmermann, Kathleen R. Stoof-Leichsenring, Stefan Kruse, Juliane Müller, Rüdiger Stein, Ralf Tiedemann, and Ullrike Herzschuh

Dear Henriette Kolling,

thank you very much for your comments and questions which helped to improve the quality of this manuscript. Please, find below your comments and questions in bold letters, while our answers are placed below and changes that will be made in the text are underlined. New references that will be included are placed at the end of this document. Line numbers are referring to changes made in the revised manuscript.

**The term "richness" is used very often, and I have difficulties understanding that term. Could you please add some information on this term, as you use it very often.**

We included the following sentence:

L149-150: "As a measure of alpha-diversity we calculated richness of (1) ASVs (number of amplicon sequence variants) and (2) unique taxonomic names (number of grouped ASVs that were assigned to the same taxonomic name)."

**Abstract: The Abstract is missing some information on the investigated material/samples.**

We included the information in the abstract.

L17: "By amplifying a short, partial *rbcL* marker on sediment core MSM05/5-712-2, […]"

**Introduction: I have the feeling other sea ice proxies should be mentioned. Further, some information on the advantages of the sedaDNA study you present here is missing. It is not clear why sedaDNA may be of advantage for sea ice reconstructions, as the sea ice biomarker IP25 is not as prone to dissolution as microfossils.**

We agree. We moved the paragraph further below after we introduced ancient DNA and have re-written it.

**Chapter 6: I feel this chapter needs more detail. The mismatch of IP25 and sedaDNA was to be expected, as you do not investigate the IP25 producers. Could you elaborate more on the reasons for the mismatch of biomarker and sedaDNA record? What about seasonality of N. frigida and IP25 production? From SW Greenland, Krawzcyk et al (2015; Polar Biology) found that N. frigida is most abundant in late winter before the main spring bloom – which is expected to be the main production season of IP25 in Fram Strait. What about habitats (under the ice, inside the ice) between the different species? And finally what are your recommendations for future work or when using sedaDN for sea-ice reconstructions?**

We have re-structured section 6 according to comments made by the other reviewers and discussed more about the ecology of *Nitzschia frigida* in this context.

At the end of our conclusions, we added the following recommendation for future work:

"Recommendations for future work with sedimentary ancient DNA in the context of sea ice reconstructions involve the preparation of reference genomes and a more targeted enrichment, for example of genes that help species to adapt to sea ice and allow them to cope with rapidly changing environmental conditions."

**L36 Krawczyk et al., 2017 should be added here**

We added the reference.

**L59 & L60 overuse of whether**

We exchanged the first occurrence with if.

**L82 I find the description of the sea ice condition in the working area confusing.**

We changes the sentence to:

"Today, the site is located south of the winter and summer sea-ice margin and is ice-free year round […]"

**L82 mentioning past sea ice variability feels wrong here, maybe add this information with some more detail to the introduction**

We moved the sentence to L64 and changed the sentence to:

"As previous work indicates variability in the past sea-ice cover (Falardeau et al., 2018; Müller et al., 2012; Müller and Stein, 2014; Werner et al., 2011, 2013), samples were chosen  according to high, medium and low concentrations of the diatom produced sea-ice biomarker $IP_{25}$ (Müller et al., 2012; Müller and Stein, 2014) and we expect associated changes in the taxonomic composition."

**L86 should say Epp et al. (2019), L116 should say Callahan et al. (2017), L120 should say Dulias et al. (2017) and Stoof-Leichsenring et al. (2012)**

We have changed them accordingly.

**L134 I do not understand this Quote.**

We changed the sentence to:

L143-144: "We resampled the dataset 100 times to the minimum number of sequences available (25,601 counts), then, for each replicate, we calculated the mean number of sequence counts for each ASV across the 100 resampling steps (code available at: https://github.com/StefanKruse/R_Rarefaction (Kruse, 2019))."

**L150-159 I feel the information on lake studies has too much detail whereas the information on marine studies is too short as the presented study is marine.**

Here we want to show, that only few studies exist that focus on ancient DNA from diatoms. As a couple of weeks ago a study about diatom and foraminiferal ancient DNA was published (Pawłowska et al., 2020) and we will add this to the list as well as a study about resurrection ecology.

We included the underlined statement, to make this clearer:

We used *sed*aDNA metabarcoding by applying the diatom-specific *rbcL_76* marker (Stoof-Leichsenring et al., 2012) which has already proved successful in low-productivity lakes of northern Siberia (Dulias et al., 2017; Stoof-Leichsenring et al., 2014, 2015), but so far has not been tested on marine sediments.

**L198 This is a major problem. However cannot be changed for your study but I welcome this comment. For future studies the parallel investigation of biomarkers and diatoms in the microfossil and genetic record may be a very promising approach.**

Yes, we agree.

---

## Author Response (AR1)

**Final response**

Dear Prof. Dr. Yuelu Jiang

We would like to thank you very much for considering our manuscript for publication in Ocean Science and the reviewers Jessica Ray (RC1), John Kirkpatrick (RC2) and Henriette Kolling (SC1) for the helpful comments and recommendations. Please, find below the revisions to our manuscript "Changes in the composition of marine and sea-ice diatoms derived from sedimentary ancient DNA of the eastern Fram Strait over the past 30,000 years". We have revised the manuscript accordingly and provide the answers to those comments (bold), our replies and the changes we made (underlined) point-by-point with line numbers referring to the revised manuscript. In general, we agree with the critiques and re-phrased unclear sections, included new statistical analyses, optimized figures as recommended and provided new supplementary material. Furthermore, the manuscript went through English correction again. As line numbers changed slightly in comparison to the Answers to the reviewer's comments, we provide here the adjusted line numbers after English correction. We hope that our revisions have improved the quality of the manuscript sufficiently for future publication in Ocean Science and are looking forward to receive your decision.

With kind regards and on behalf of the co-authors

Heike Zimmermann

**Answers to os-2019-113_RC1**

**l.127 - Less than 50% of 3 replicates?**

We agree that this is confusing and changed the sentence from:

"[…](3) were present at least 3 times among the different replicates, (4) showed taxonomic resolution below the phylum level "Bacillariophyta" and (5) were tagged as "internal" by obiclean in less than 50 % of the different replicates to reduce PCR and sequencing artefacts."

L145-147:

"[…](3) were present at least 3 times among all sequenced PCR products, (4) showed taxonomic resolution below the phylum level "Bacillariophyta" and (5) were tagged as "internal" by obiclean in less than 50 % of all sequenced PCR products to reduce PCR and sequencing artefacts."

**l.147 - What was the constraining factor used for CONISS analysis? Please include.**

The constraining factor was depth. With regard to your concerns below we decided to omit the CONISS analysis.

**l.201 - What are the rare ASVs? A quick perusal of the Table S1 indicates many zeros. Would it be possible or useful to include a supplementary figure showing rank-abundance curves, either for the entire dataset or for the individual samples?**

The majority of ASVs are rare (less than 1% per sample). We will provide an additional supplemental file including the rank abundance curves for each depth (see preliminary Fig. 1 below). We decided to also included several stratigraphic diagrams containing the ASVs without grouping as **Suppl. Fig. S1.**

L232-234: The replicates of each sample show some variations (Fig. 3) in the presence and abundance of ASVs, especially for ASVs amounting to less than 1 % per sample (Suppl. Table 2, **Suppl. Fig S2**).

[Figure]

*Figure 1: Rank-abundance plot for each sample (depth in m) based on amplicon sequence variants (ASVs).*

**l.205 - Interesting theory, but why would inhibition only appear in some samples? I think this might be easy to test by a dilution PCR of a few representative samples. Also, if there are differences in the relative amount of diatom DNA in each sedaDNA sample, would it be possible to do a quick qPCR check of rbcL target abundance using the same primer set?**

Unfortunately, we could not go back to the lab as work there was reduced to a minimum the past months. We discussed this and decided that it is most likely due to different amounts of diatom template molecules and deleted the part referring to inhibition.

L242-243: It is possible that higher dissimilarities between some replicates are the result of the low amount of template DNA .

**l.209 (Section 5) - My first impression from reading the results is that the authors have some difficulty in interpreting diatom community assemblage differences according to the results of the CONISS analysis, i.e. division into five aggregate "zones". In particular because trends in the relative abundance of specific sympagic diatom taxa between different CONISS "zones" are mentioned but not statistically tested. This leads me to question whether CONISS analysis is useful given the present dataset. Were any other analyses attempted to identify discriminant ASVs/taxa? And how did the authors conclude that there are two diatom assemblage reorganizations (l.340) when the CONISS analysis identifies four?**

We agree, that this method might not be ideal and removed the CONISS analysis. We have deleted all references to this method and thus slightly adjusted the text.

We also changed the sentences in the abstract and conclusion with this regard:

L21-23: "Taxonomic composition is dominated by cold-water and sea-ice associated diatoms and suggests several re-organizations –after the Last Glacial Maximum, Younger Dryas, after the Early and after the Mid-Holocene."

L402-404: "The *sed*aDNA record captures substantial temporal change of diatom taxonomic composition and richness with four compositional re-organizations: the first between 20.51 and 15.1 cal kyr BP, the second between 12.8 and 10.1 cal kyr BP, the third between 10.1 and 5 cal kyr BP and the fourth between 5 and 1.6 cal kyr BP."

**l.223 - Could you please elaborate on what is meant by "richness of taxonomic names". According to l.184 different ASVs can be assigned to the same name, so how might this affect the apparent richness? And why use "taxonomic names" for richness calculations when taxonomic rank assignment is not uniform across all ASVs?**

With "richness of taxonomic names" we mean grouped ASVs assigned to the same taxonomic name. We compared rarefied taxonomic richness based on (1) ASVs and (2) also for grouped ASVs assigned to the same taxonomic name. While changes of ASV-derived richness vary stronger between the samples in comparison to the changes of ASVs grouped by their taxonomic name, the trends are similar: Richness is highest in the last glacial samples, lower in the deglacial samples and lowest in the Holocene samples.

L258-259: We changed the sentence to:

"Generally, the richness of both ASVs and unique taxonomic names (ASVs grouped based on identically assigned taxonomic names) is higher in samples dated to the last glacial in comparison to those dated to the Holocene (Fig 4)."

Furthermore. we changed the captions of Fig. 4 and 5 to:

"Figure 4: Barplots showing the rarefied (a) number of amplicon sequence variants (ASVs) per sample and (b) number of unique taxonomic names grouped ASVs assigned to the same taxonomic name for each sample with depth (m) of the sediment core MSM05/5-712-2."

"Figure 2: Taxonomic composition with relative proportions (%) of the 360 detected sequence variants grouped into 75 unique taxonomic names based on identically assigned taxonomic names of sediment core MSM05/5-712-2. Taxonomic names are sorted according to the weighted average with depth."

**l.224 - I am not quite comfortable with the use of "turnover" in the context it is used, since the samples are discrete and therefore a discontinuous representation of time. In my opinion, "turnover" suggests a biological/ecological linkage from one sample to the next, while in this study the samples compared are isolated snapshots in time. Can the authors comment on the choice to use this term?**

We agree, that the term is misleading. We chose the term as we see a distinct change in the taxonomic composition, e.g. after the LGM. But since we did not calculate beta-diversity (exactly because we only have a few samples) we changed "turnover" to "shift" in all occurrences.

**Fig.5 - I wonder if the reader might find this figure somewhat difficult to interpret given that relative taxon abundance at multiple taxonomic ranks are presented for each sample. Could the authors please the reasoning for presenting the data in this way?**

Stratigraphic diagrams are a common representation in paleoecology, for example for pollen or microfossil analysis. As taxonomic resolution differs in different families or genera (for example due to lack of morphological differences or here due to sequence similarity of closely related species in the databse), showing the detected organisms on the lowermost taxonomic level as possible was our goal. Out intentions was to used wa.order="topleft" option in strat.plot, which sorts the taxa according to the weighted average with depth to better visualize the change over time. However, we see your point and will omit the weighted average and make the plot more informative, e.g. by separating centrics from pennates. Again, in the manuscript we want to show the grouped version to make the plot as complete as possible. However, we made a new supplementary file with several figures showing strat.plots of all 360 ASVs.

[Figure]

**Figure 5:** Stratigraphic diagrams with sequences assigned to (a) pennate diatoms (blue) and (b) centric diatoms (green) and higher level sequences assigned between family and phylum level (gray) The taxonomic composition with relative proportions (%) of the 360 detected sequence variants is grouped into 75 unique taxonomic names based on identically assigned taxonomic names of sediment core MSM05/5-712-2.

**Table 1 - The Paleoenvironmental conditions descriptions seem somewhat arbitrary. For example, how is "sea-ice retreat" (3.1 m depth) different from "Reduced sea-ice cover allowing spring sea-ice algal and summer phytoplankton productivity" (7.85 m depth).**

We agree. We cited the descriptions given in the original research. We decided to omit the table.

**l.239 – foraminifer**

Changed. Thank you.

**Fig.6 - Family-level taxonomy is presented but l.170-174 states that this collation may mask functional differences.**

We provided stratigraphic diagrams for all 360 ASVs in the supplement. This is figure was supposed to give an alternative representation of the data on family level in context with $IP_{25}$. We also sorted the taxa according to centrics and pennates, so that the figure will be easier to understand.

[Figure]

**Fig.6 - The double top axes (depth and age) are very helpful, but identify very clear differences in sedimentation rates in the downcore. For example, zone II has higher temporal resolution than zone I. Might not differences in sedimentation rates also affect sedaDNA signal?**

Yes, a higher temporal resolution indeed could affect the sedaDNA signal. This could be an additional explanation for the lower richness of zone II in comparison to zone I. However, the distinct shift we see in taxonomic composition is rather an environmental signal. We added the following sentence.

L299-301: "Higher sedimentation rates and thus higher temporal resolution during the Bølling/Allerød phase could have affected the *sed*aDNA signal, yet the distinct shift in taxonomic composition suggests, that this is rather an effect of the changing environmental conditions during this phase."

**l.247 - Again, ambiguous results from CONISS analysis?**

Changed

**l.336 - "highly detailed taxonomic resolution" depends on what is meant by highly detailed, and what fraction of the data is being referred to. I suggest moderating this statement.**

L398: We deleted the word "highly". We mention in this sentence that we refer to the fraction of data resolved on genus and species level. In total 64.4% of our 360 ASVs of the filtered dataset are resolved on genus or species level (see line 163), and we believe this is quite detailed. But

**l.320 - Very interesting that N. cf. frigida sticks out as a possible new sea-ice proxy. However, according to Fig.6, this taxon as highest relative abundance in the most recent sample (0.4 kya BP) when sea-ice cover is low. What about Cylindrotheca closterium? Or Haslea avium? Again, I think it would be very helpful if taxon relative abundances were statistically tested in order to identify ASVs/taxa that contribute significantly the observed diatom diversity in different samples.**

Around 0.4 kyr BP, which falls in the temporal phase of the Little Ice Age, IP25 values are increasing. So there is sea ice influence. Results of our statistical analysis can be found in the comments of Reviewer 2. We did not find a linear relationship between any of the ASV in our record and $IP_{25}$. However, we have re-written the section (L357-398) and included the new results.

**Answers to os-2019-113_RC2**

**Methods: Section 2.1: The coring equipment and handling should be described. Was this a piston core? What diameter? How was it handled? Were the cores halved onboard? Did the authors use an archive half or a working half that had previously been used for other sampling activities? Most importantly, how was the core stored – at what temperature? If at -80 C, this should be clearly stated. If not at -80 C, it should be acknowledged that DNA preservation ex situ may be imperfect.**

The core was taken in 2007 with a Kastenlot corer (gravity) that has a diameter of 30 x 30 cm. As this is sufficient material, about 1m long subcores are taken and stored at 4°C. The core has to be opened and subsampled on board. However, the subcores had also been used for other sampling activities. We will refer to the supplement of Gersonde et al. (2012) in the manuscript where the procedure is explained in detail.

We will change the sentence to:

L88-91: "The kastenlot core MSM05/5-712-2 (N 78.915662, E 6.767167, water depth 1487 m) was collected from the western continental slope of Svalbard during the cruise of Maria S. Merian (Budéus, 2007) in the eastern Fram Strait in 2007 (Fig. 1). On board, subsections of 1 m length are taken in square plastic boxes as explained in the supplement of Gersonde et al. (2012) and stored at 4°C. Therefore, DNA preservation might be imperfect."

The last sentence changed due to English correction to:

"On board, subsections of 1 m length were placed in square plastic boxes as explained in the supplement of Gersonde et al. (2012) and stored at 4°C. This may have affected DNA preservation."

**Do the authors have any tracers or controls for post-coring seawater influence on the sediment core? I am willing to believe that the authors, working in a dedicated lab, were meticulous about their sampling and extraction. Did they use tracer DNA as described by Epp et al.?**

At the time of coring (2007), it was not planned to analyze ancient DNA. Hence, no tracers were added. We understand such concerns, but the chance of post-coring contamination deep inside a Kastenlot core seems quite low. A supporting argument is that if we had such contamination, we would not see distinct changes among our samples that are consistent with past climatic changes.

**Section 2.2: This is the most significant concern I have about this dataset. The primer set used was designed by Dulias et al. for freshwater lakes. Its specificity for diatoms appears to be very high, as the authors highlight, which is good to address. However, it contains multiple mismatches on both the forward and reverse primers for the taxa noted to contribute to the IP25 proxy (Haslea, Pleurosigma; please see attached "supplement"). (I did not look at other marine taxa, but that should be investigated.)**

**Please also note the supplement to this comment:**

https://www.ocean-sci-discuss.net/os-2019-113/os-2019-113-RC2-supplement.pdf

The primers were designed to function as a general diatom-specific marker by Stoof-Leichsenring et al. (2012), but since then they were mostly applied on lacustrine sediments. We added this information in the introduction:

L78-80: "We used sedaDNA metabarcoding by applying the diatom-specific *rbcL_76* marker (Stoof-Leichsenring et al., 2012) which has already proved successful in low-productivity lakes of northern Siberia (Dulias et al., 2017; Stoof-Leichsenring et al., 2014, 2015), but so far has not been tested on marine sediments."

However, these primers were not tagged for parallel high-throughput sequencing. Tagging and adjustment of PCR conditions was performed by Dulias et al. (2017). We therefore added this information in the methods section:

L118-120: "The PCR reaction mixes and conditions were prepared following the adjusted protocol for tagged *Diat_rbcL_705F* and *Diat_rbcL_808R* primers as described in Dulias et al. (2017) with the exception that 3 µl DNA (DNA concentration 3 ng µL$^{-1}$) was used as a template."

We agree with your concern. Mismatches in primer binding sites are not ideal, but using DNA metabarcoding always comes with a trade-off. The primer binding sites were chosen to maximize (1) amplification success by having the amplicon short, (2) retrieving as many diatom (pennates and centrics) species as possible with the best resolution possible and (3) it should co-amplify as few non-diatom groups as possible.

Only 3 species are known to produce IP$_{25}$: *Pleurosigma stuxbergii* var. *rhomboides*, *Haslea spicula* and *Haslea kjellmanii* (Limoges et al., 2018). Of those, only *Pleurosigma stuxbergii* has a reference sequence containing our marker and we detected only 1 mismatch in the forward and one mismatch in the reverse primer-binding site. *Haslea nusanatra* and *Haslea howeana* which you provided in your supplement have a tropical distribution. On the other hand, *Haslea avium*, which was also detected in our study, shows only 1 mismatch in the forward and one mismatch in the reverse primer-binding site. We checked the number of mismatches in the forward and the reverse primers with the 15 *Haslea* spp. reference sequences in our database and found between 1-5 mismatched in forward and 0-2 mismatches

in reverse primers. The number of mismatches *Haslea spicula* and *Haslea kjellmanii* have is certainly in that range. However, most sequences that we retrieve show mismatches: e.g. *Thalassiosira* species mostly have 2-3 mismatches with the forward and 0-2 with the reverse primers. And up to 5 mismatches with the forward and 1 mismatch with the reverse primer can be found for *Chaetoceros socialis* (assigned to GenBank Accession FJ002154).

**As a result this dataset should not be considered to be a well-rounded assessment of diatoms in this setting. Any of the increases / decreases in richness, or the absence of certain taxa, or overall assessment of diatom diversity are not valid as we are effectively blinded to many of the taxa known to be present in this location (and presumably throughout the sediment record).**

This concern is based on the assumption that the primers were specifically designed for freshwater lakes, which they were not. We agree that absences in our data does not necessarily mean absence in the past.

We included the following sentence:

L230: "The recovery of taxa by *sed*aDNA metabarcoding is prone to false presences or absences. As our study lacks a morphological diatom record, false absences cannot be assessed and true presences cannot be confirmed, which means that absence in our record does not necessarily translate to a true physical absence in the past."

**This could perhaps be partially addressed by compiling an alignment of diatom chloroplast rbcL sequences documented to be or have been present at this location and noting how many taxa you would expect to find do or don't have mismatches with the primers. Maybe the authors are aware of this (?). At the end of the day, the detection of the sequences found is real data, and valuable. However, a broad assessment of shifts in diversity and community responses to climate is very problematic.**

We agree with your concern. Unfortunately we only know a fraction of the species that occur(red) in the area. The region is overall understudied by molecular surveys and dissolution leads to incomplete microfossil records. Hence, we feel that we cannot provide such an alignment, as it would be incomplete as well. About the mismatches, please have a look 2 questions above. This is a known issue and wobble bases are included in the primers for mitigation of this bias. If it is desired, we can prepare a figure to visualize the primer mismatches.

**On a different note, the cycle numbers and annealing temperatures, in my opinion, should be included even if they are included in the given citation. 50 cycles is a very large number. Did the authors ever visualize bands in their extraction or PCR negative controls?**

We agree, that 50 cycles are a very large number, but large cycle numbers are not uncommon in ancient DNA studies (up to 65 cycles: (Willerslev et al., 2014) or 45 cycles: (Voldstad et al., 2020)). We point these issues out in lines 210-214. For future studies, reducing cycle numbers to 45 is planned, but we cannot change this for the current project. However, reducing cycle numbers also comes with a trade-off, as this has shown to reduce replicability (Krehenwinkel et al., 2017; Nichols et al., 2018).

L214: We included the reference Nichols et al., 2018.

Yes, after each PCR, we first evaluated bands on an agarose gel. We will include the following sentences:

L120-123: "PCRs were performed with the following settings: 5 minutes at 94°C (initial denaturation), then 50 cycles at 94°C (denaturation), 49°C (annealing) and 68°C (elongation) and a final elongation step at 72°C for 5 minutes. Subsequently the PCR success was checked with gel-electrophoresis."

**Line 110 – what version (chemistry)?**

We change the sentence to:

L126: "The sequencing library was prepared with the Mid Output kit v 2 according to the Fasteris Metafast protocol for low complexity […]"

**Section 2.3: I'm OK with the use of ASVs and I appreciate the authors taking time to explain why they didn't use OTUs. Regarding their reference database, how many references did this method produce? Did these references include diatoms documented in other studies from the waters around Svalbard? The reference database is a frequent scapegoat in the discussion; release 138 is a couple years ago. The taxonomy seems pretty robust to me but it could be updated if need be.**

This method produced 2,320 references, and yes, it contained many taxa which are documented in waters around Svalbard, such as *Nitzschia frigida*, *Chaetoceros socialis*, *Thalassiosira antarctica* or *Thalassiosira nordenskioeldii* (to name a few that were also detected in our record).

The ENA/EMBL release 138 was released in November 2018, and thus up-to-date when we started the analyses. The reviewer probably confused this with the GenBank release 138 from 2003.

We changed the sentence to:

L137-139: To generate the reference database for the taxonomic assignment of the sequences we downloaded the EMBL release 138 (released November 2018) and used *ecopcr* (Ficetola et al., 2010) according to the descriptions of Dulias et al. (2017) and Stoof-Leichsenring et al. (2012) containing 2,320 reference sequences.

**Going back to the negative controls, on line 122 how many exactly is "the majority" that were singletons?**

We provide an additional Supplementary table which contain the negative controls from extractions and PCRs. Furthermore, we will change the sentence to:

L141-142: "Of the 204 different sequence variants detected in extraction and PCR negative controls 83% of their occurrences were singletons and […]"

**On line 125, why 10 read counts? How many sequences, and how many ASVs, were removed out of the total using these criteria? As the supplemental table only has the kept sequences, one can't tell.**

This is only a first threshold we use for denoising so we do not lose too much information in the beginning. We included the following information in the text:

L148: "Filtering with R reduced the number of read counts from 7,536,449 to 6,199,984."

**Section 2.4 (and throughout): Please make sure to always use the term "PCR replicate" or "PCR replication", as these are not sample replicates. PCR replication is useful but the terminology**

**should be clear as later on it may get confusing for readers who aren't methods geeks and think that these are sample or extraction replicates.**

Thank you for pointing this out. We changed this accordingly throughout the manuscript and figure captions.

**Section 2.5: Resampling 100 times might be overkill but I guess it can't hurt. The new minimum number of sequence counts doesn't quite make sense to me through – can you clarify "according to descriptions in the preceding paragraph"? (It's not quite 12 \*25,601.)**

We change the following sentence "For taxonomic composition and richness calculations, we first summed up the sequence counts for each ASV of the replicates belonging to the same sample…"

To:

L164-167: "For taxonomic composition and richness calculations, we combined the PCR replicates of the corresponding sample. This resulted in a new minimum number of sequence counts (300,415 counts) that was used for resampling the dataset 100 times according to descriptions in the preceding paragraph."

**Section 3: Line 157 – What do you mean by "outperformed"? Are you referring to the specificity for diatoms? Please clarify.**

Thank you for pointing that out. We changed the sentence to:

L187-188: "Possibly owing to a much shorter marker size, the *rbcL_76* marker also surpassed the usually amplified 18S rDNA markers with regard to specificity for diatoms (Coolen et al., 2007; De Schepper et al., 2019; Kirkpatrick et al., 2016)"

**Line 164 – "striking proportion" – I don't think it's that striking, in fact, 4.8% seems pretty low!**

L194: We deleted the word striking.

**Line 177 – can you give a numeric range for the copy number variation in published work?**

Unfortunately we cannot give a numeric range, as most papers we found mention this as vaguely as we did. However, we included another reference, where the authors showed that rbcL copy number was significantly different in the 8 tested benthic freshwater diatoms and that this variation is positively correlated with cell biovolume (Vasselon et al., 2018).

We changed the sentence to:

L208-210: "Furthermore, the enrichment of centric diatoms in the *sed*aDNA record could be the result of copy number variation of the *rbcL* gene between different species and cell biovolume (Bedoshvili et al., 2009; Round et al., 1990; Vasselon et al., 2018)."

L81: We also included the reference (Vasselon et al., 2018) here.

**Section 4: Here I think again it is important to be clear that the replicates being discussed are PCR replicates of the same template DNA. Also, what happened to the two sample replicates from the deepest sample? How similar or dissimilar were they and does the sample variation exceed the variation found in PCR replication? While two duplicates aren't statistically as useful as a triplicate, they need to be discussed in the context of reproducibility. Doing so would, I think,**

**strengthen the authors' arguments. This should be addressed in revision (unless I missed it somewhere?).**

We already handled PCR replicates, as mentioned above.

In the NMDS plot (see below the improved version of the figure), you can see, that the 3 PCR replicates of the two samples at 8.85 m depth cluster so narrowly, that it is really hard to identify the labels at all. Hence, they were very similar. However, as this is an ordination biplot, we cannot move around the labels. Therefore, we included the colours of the polygons in the caption and hope that this makes it clearer.

The degree of dissimilarity is visualized in the NMDS plot. We agree that more replicates always help to decrease the level of uncertainty and include this statement in the text:

L238-240: "For the oldest sample at 8.85 m depth we additionally processed a sample replicate. The PCR replicates of both sediment samples at 8.85 m depth were highly similar and clustered tightly together in the NMDS plot (Fig. 3). Although a higher number of replicates would improve the robustness of our analysis [...]"

We changed the following sentence to make it clear, that the PCR replicates are based on the same template:

L 232-233: "The PCR replicates (different PCR products from the same DNA extract) of each sample show some variations (Fig. 3) [...]"

[Figure]

Figure 3: Non-metric multidimensional scaling plot based on the filtered and resampled diatom dataset with the PCR replicates (indicated by an underscore with a number) of a sample (depth in m) linked in a polygon of sample-specific colour: light green = 0.12m, green = 0.3m, light violet = 0.9 m, violet =

3.1 m, light blue = 3.71 m, dark blue = 5.8 m, turquoise = 6.68 m, grey =7.27 m, red = 7. 85 m, brown = 8.56 m, orange = 8.85 m (2 sample replicates with 3 PCR-replicates each).

**Section 5: Lines 228-299: This inverse relationship, if significant, is intriguing! Please test statistically for significance.**

The inverse relationship between *Chaetoceros* and *Thalassiosira* was tested for significance on family level. The Pearson's correlation efficient is -0.61126971, with a p-value of 0.045701848. Hence, there is a moderate significant negative correlation. We included the information in brackets into the sentence:

L264: "A general trend that can also be observed at the family level is an inverse relationship (r=-0.61, p= 0.046) of the dominant families Thalassiosiraceae and Chaetoceraceae (Fig 6)"

Furthermore, we will include a sentence into the methods section (see answer after the comment regarding section 6).

**Lines 242-244... etc: This sort of speculation about how ice conditions may have affected diatom community diversity, etc., is undermined by the primer issue.**

Is this concern based on the assumption that the primers were specifically designed for freshwater lakes (which they were not)? Based on the sedaDNA composition (both sympagic and pelagic diatoms were detected) and by taking into consideration the results of other proxies during this time, it is conceivable that conditions were heterogeneous throughout the seasons. We changed the sentence slightly:

L279-282: "It is conceivable that a heterogeneous and dynamic environment produced by winter sea-ice cover with ice-free conditions during summer probably allowed for diverse diatom communities to develop in the different habitats and over the seasons, which is reflected in the highest numbers overall of ASVs in samples dated to this time span."

Due to English correction the sentence changed to:

"It is conceivable that a heterogeneous and dynamic environment produced by winter sea-ice cover with ice-free conditions during summer could allow diverse diatom communities to develop in the different habitats and over the seasons, which is suggested by the highest overall numbers of ASVs in samples dated to this time span."

**Lines 253 and elsewhere: "Low proportions of Nitzchia cf. frigida..." and similar statements should be reworded, as we do not know the proportion of diatoms. We know the proportions of the sequences found in the dataset after extensive amplification (50 cycles!). Please be careful to keep this distinction clear.**

We included the proportions in brackets after each statement.

**Line 265: How far away is that (Hinlopen Strait)? In this section in general, there are numerous and interesting comparisons to published records from related locations. It would be appreciated if the proximity of these (in km) were clear for the less-familiar reader. (E.g. also line 276, 278, 291, 304, 323)**

We added an annotation of the places mentioned in our text into the map.

[Figure]

**Line 274: "peak proportions", "Lower proportions" – can you please quantify. (Also e.g. line 301.)**

Yes, we included the proportions in brackets after each statement.

**Line 289: Richness and diversity are not the same; please be specific.**

We will consistently refer to richness instead of diversity as recommended.

**Section 6: Beyond the methods concerns, there are still some interesting points here. These could be strengthened by a statistical analysis testing the correlation between the new data to the published IP25 proxy (lines 319 - 323).**

We have tested for significance, first on family level and second on ASV level and will add this information in the text, the plots showing significantly correlated families or ASVs (see Figure below) will be added in the Supplement. As we had to interpolate $IP_{25}$ values for the depths that were analysed in this study, we include a paragraph describing the process in the methods section. We plotted the significantly correlated ASVs (see table below and figure below) and found that they do not show a linear relationship with $IP_{25}$, which is why we will not include the table in the manuscript. However, some of the patterns are quite interesting (see figures below the table) and we have re-structured section 6 and included new results and parts of the discussion about primer mismatches in Haslea and Pleurosigma.

We included:

L174-179: "For correlation analysis we interpolated $IP_{25}$ values using the methods described in Reschke et al. (Reschke et al., 2019). Therefore, the $IP_{25}$ data were transformed using function *zoo* from the "zoo" package (Zeileis and Grothendieck, 2005) and used in the function *CorIrregTimser* using the package "corit" (https://github.com/EarthSystemDiagnostics/corit). The correlation between Chaetocerotaceae and Thalassiosiraceae as well as between $IP_{25}$ and all ASVs was tested for significance using R function

*rcorr* (method Pearson) from the package "Hmisc" (Hollander and Wolfe, 1975; Press et al., 1988)(Quelle)."

**Table 1: Amplicon sequence variants (ASVs) with significant (p-value) correlation.**

| ASV | Correlation coefficient | p-value |
| --- | --- | --- |
| Attheya28 | 0.7542327 | 0.00732029 |
| Bacillariaceae71 | 0.6970434 | 0.01713238 |
| Bacillariophyta157 | 0.7159116 | 0.01322211 |
| Bacillariophyta245 | -0.6217257 | 0.0411348 |
| Chaetoceros.cf.pseudobrevis.1.SEH.2013461 | 0.6619875 | 0.02649079 |
| Cylindrotheca.closterium558 | 0.6387489 | 0.03439247 |
| Gomphonema586 | -0.6077406 | 0.04731908 |
| Haslea.avium589 | 0.6899115 | 0.01880586 |
| Nitzschia.cf.frigida709 | 0.8093591 | 0.0025435 |
| Thalassiosira1017 | -0.673798 | 0.02301182 |

The following figure and caption are included as additional supplement.

[Figure]

**Figure S3: Amplicon sequence variants with significant correlation to interpolated values of the sea ice biomarker IP$_{25}$ with loess smoothing (blue line) and standard error (dark grey area).**

**Technical corrections: Figure 3: Labels are not readable. Figure 5: This appears upside-down. More importantly, the labels and proportion bars are very, very small; but much of the graph is empty whitespace. Please try making the bars wider. Also, consider organizing the taxa by e.g. rank order (based on proportion) and breaking into 2 or 3 separate figures. Figure 56: Please define "higher level" in the caption.**

Fig. 3: As we don't want to manipulate the biplot by spreading apart the labels, we attributed distinct colours and will refer to PCR replicates in the caption. We placed the figure to another answer further above (Figure 3).

Fig. 5: Yes, we split the graph into 2 graphs and included further recommendations from Jessica Ray (Reviewer1). Originally we used a rank order (based on proportion): wa.order="topleft" option in strat.plot, which sorts the taxa according to the weighted average with depth.

We defined higher level in the caption as: "Higher level contains sequences assigned between family and phylum level."

**Answers to os-2019-113_SC1**

**The term "richness" is used very often, and I have difficulties understanding that term. Could you please add some information on this term, as you use it very often.**

We included the following sentence:

L162-163: "As a measure of alpha-diversity we calculated richness of (1) ASVs (number of amplicon sequence variants) and (2) unique taxonomic names (number of grouped ASVs that were assigned to the same taxonomic name)."

**Abstract: The Abstract is missing some information on the investigated material/samples.**

We included the information in the abstract.

L17-18: "By amplifying a short, partial *rbcL* marker on sediment core MSM05/5-712-2, […]"

**Introduction: I have the feeling other sea ice proxies should be mentioned. Further, some information on the advantages of the sedaDNA study you present here is missing. It is not clear why sedaDNA may be of advantage for sea ice reconstructions, as the sea ice biomarker IP25 is not as prone to dissolution as microfossils.**

We agree. We moved the paragraph further below after we introduced ancient DNA and have re-written it.

L42-45: "Next to microfossil-based reconstructions, the diatom produced sea-ice proxy IP25 (a highly branched isoprenoid alkene with 25 carbon atoms; (Belt et al., 2007)) combined with phytoplankton biomarkers (e.g. brassicasterol, dinosterol; (Volkman, 1986)) permit semi-quantitative reconstructions of past sea-ice distribution (Belt, 2018; Belt and Müller, 2013; Müller et al., 2009; Müller and Stein, 2014; Stein et al., 2012, 2017). However, diatoms […]"

**Chapter 6: I feel this chapter needs more detail. The mismatch of IP25 and sedaDNA was to be expected, as you do not investigate the IP25 producers. Could you elaborate more on the reasons for the mismatch of biomarker and sedaDNA record? What about seasonality of N. frigida and IP25 production? From SW Greenland, Krawzcyk et al (2015; Polar Biology) found that N. frigida is most abundant in late winter before the main spring bloom – which is expected to be the main production season of IP25 in Fram Strait. What about habitats (under the ice, inside the ice) between the different species? And finally what are your recommendations for future work or when using sedaDN for sea-ice reconstructions?**

We have re-structured section 6 according to comments made by the other reviewers and discussed more about the ecology of *Nitzschia frigida* in this context.

At the end of our conclusions, we added the following recommendation for future work:

L407-410:"Recommendations for future work with sedimentary ancient DNA in the context of sea ice reconstructions include the preparation of reference genomes and a more targeted enrichment, for example of genes that help species to adapt to sea ice and allow them to cope with rapidly changing environmental conditions."

**L36 Krawczyk et al., 2017 should be added here**

L41 + L382: We added the reference.

**L59 & L60 overuse of whether**

L69: We exchanged the first occurrence with if.

**L82 I find the description of the sea ice condition in the working area confusing.**

We changes the sentence to:

L96: "Today, the site is located south of the winter and summer sea-ice margin and is ice-free year round […]"

**L82 mentioning past sea ice variability feels wrong here, maybe add this information with some more detail to the introduction**

We moved the sentence to L74-76 and changed the sentence to:

"As previous work indicates variability in the past sea-ice cover (Falardeau et al., 2018; Müller et al., 2012; Müller and Stein, 2014; Werner et al., 2011, 2013), samples were chosen according to high, medium, and low concentrations of the diatom produced sea-ice biomarker IP$_{25}$ (Müller et al., 2012; Müller and Stein, 2014) and we expect associated changes in the taxonomic composition."

**L86 should say Epp et al. (2019), L116 should say Callahan et al. (2017), L120 should say Dulias et al. (2017) and Stoof-Leichsenring et al. (2012)**

We have changed them accordingly.

**L134 I do not understand this Quote.**

We changed the sentence to:

L154-157: "We resampled the dataset 100 times to the minimum number of sequences available (25,601 counts), then, for each replicate, we calculated the mean number of sequence counts for each ASV across the 100 resampling steps (code available at: https://github.com/StefanKruse/R_Rarefaction; Kruse, 2019)."

**L150-159 I feel the information on lake studies has too much detail whereas the information on marine studies is too short as the presented study is marine.**

Here we want to show, that only few studies exist that focus on ancient DNA from diatoms. As a couple of weeks ago a study about diatom and foraminiferal ancient DNA was published (Pawłowska et al., 2020) and we will add this to the list at

L68: […] or diatoms in particular (Pawłowska et al., 2020).

We included the underlined statement, to make this clearer:

L78-80: We used *sed*aDNA metabarcoding by applying the diatom-specific *rbcL_76* marker (Stoof-Leichsenring et al., 2012) which has already proved successful in low-productivity lakes of northern Siberia (Dulias et al., 2017; Stoof-Leichsenring et al., 2014, 2015), but so far has not been tested on marine sediments.

**L198 This is a major problem. However cannot be changed for your study but I welcome this comment. For future studies the parallel investigation of biomarkers and diatoms in the microfossil and genetic record may be a very promising approach.**

Yes, we agree.

**Further corrections we made:**

1. L4-5: Rüdiger Stein would like his name spelled Ruediger in the author list.
2. We adjusted all µl to µL and replaced division signs such as g/ µL with g µL$^{-1}$
3. L140: As we prepared Suppl. Table 2, we found that the last column was not included in building the sum of extraction negative controls and we changed it accordingly from 201 to 237 counts.
4. In the comments of SC1 there was some critique about the introduction and we felt, we have to improve the first paragraph of the introduction.

   L28-33:We changed it from:
   "The global climate system is strongly coupled with Arctic sea-ice cover (Goosse and Fichefet, 1999): yet Earth system models display large uncertainties in projections of sea ice, which stresses the need for paleo sea-ice reconstructions that can be used to improve the models (de Vernal et al., 2013). Currently, semi-quantitative reconstructions of past sea-ice distribution can be achieved by combining the diatom produced sea-ice proxy IP$_{25}$ (a highly branched isoprenoid alkene with 25 carbon atoms; (Belt et al., 2007)) and phytoplankton biomarkers (e.g. brassicasterol, dinosterol; (Volkman, 1986)) detected in down-core sediments ."

   to:
   L33-39: "The marine environment is a complex ecosystem in which the distribution of organisms is controlled significantly by abiotic constraints such as sea-surface temperatures (SSTs), salinity, nutrient distribution, light conditions, and sea-ice cover (Cherkasheva et al., 2014; Ibarbalz et al., 2019; Nöthig et al., 2015; Pierella Karlusich et al., 2020). Over the past 30,000 years the subarctic North Atlantic Ocean was subject to frequent sea-ice expansions and contractions (Müller et al., 2009; Müller and Stein, 2014; Syring et al., 2020; Werner et al., 2013), which are expected to have affected the composition of the regional species pool. Diatoms (Bacillariophyta) are unicellular, siliceous organisms that are photoautotrophic and thrive in the euphotic zone of the ocean."
5. L428: We included a sentence to the Acknowledgements:

"We thank Cathy Jenks for English correction."

6. We found two small mistakes in the references and corrected them:

L504-505: Fetterer, F., Knowles, K., Meier, W. N., Savoie, M. and Windnagel, A. K.: Sea Ice Index, Version 3. Monthly and daily GIS compatible shapefiles of median ice extent, National Snow & Ice Data Center, doi:10.7265/n5k072f8, 2017.

L518-519: Harrison, W. G. and Cota, G. F.: Primary production in polar waters: relation to nutrient availability, Polar Research, 10(1), 87–104, doi:10.3402/polar.v10i1.6730, 1991.

**References:**

[revised manuscript text omitted]

Willerslev, E., Davison, J., Moora, M., Zobel, M., Coissac, E., Edwards, M. E., Lorenzen, E. D., Vestergard, M., Gussarova, G., Haile, J., Craine, J., Gielly, L., Boessenkool, S., Epp, L. S., Pearman, P. B., Cheddadi, R., Murray, D., Brathen, K. A., Yoccoz, N., Binney, H., Cruaud, C., Wincker, P., Goslar, T., Alsos, I. G., Bellemain, E., Brysting, A. K., Elven, R., Sonstebo, J. H., Murton, J., Sher, A., Rasmussen, M., Ronn, R., Mourier, T., Cooper, A., Austin, J., Moller, P., Froese, D., Zazula, G., Pompanon, F., Rioux, D., Niderkorn, V., Tikhonov, A., Savvinov, G., Roberts, R. G., MacPhee, R. D. E., Gilbert, M. T. P., Kjaer, K. H., Orlando, L., Brochmann, C. and Taberlet, P.: Fifty thousand years of Arctic vegetation and megafaunal diet, Nature, 506(7486), 47–51, doi:10.1038/nature12921, 2014.

[revised manuscript text omitted]